# The elevated expression of ORF75, a KSHV lytic gene, in Kaposi sarcoma lesions is driven by a GC-rich DNA *cis* element in its promoter region

**Ashwin Nair**[1], **David A. Davis**[1], **Andrew Warner**[2], **Baktiar Karim**[2], **Ramya Ramaswami**[1], **Robert Yarchoan**[1]*

**1** HIV and AIDS Malignancy Branch, Center for Cancer Research, National Cancer Institute, Bethesda, Maryland, United States of America, **2** Frederick National Laboratory, National Cancer Institute, Frederick, Maryland, United States of America

* robert.yarchoan@nih.gov

## Abstract

The spindle cells of Kaposi sarcoma (KS) lesions primarily express Kaposi sarcoma herpesvirus (KSHV) latent genes with minimal expression of lytic genes. However, recent transcriptome analyses of KS lesions have shown high expression of KSHV open reading frame (ORF) 75, which is considered a late lytic gene based on analyses in primary effusion lymphoma (PEL) lines. ORF75 encodes a pseudo-amidotransferase that is part of the viral tegument, acts as a suppressor of innate immunity, and is essential for viral lytic replication. We assessed a representative KS lesion by RNAscope and found that ORF75 RNA was expressed in the majority of latency-associated nuclear antigen (LANA)-expressing cells. Luciferase fusion reporter constructs of the ORF75 promoter were analyzed for factors potentially driving its expression in KS. The ORF75 promoter construct showed high basal transcriptional activity *in vitro* in endothelial cells, mediated by a proximal consensus specificity protein 1 (Sp1) (GGGGCGGGGC) element along with two distal CCAAT boxes. Sp proteins formed complexes with the proximal consensus Sp1 element to activate ORF75 promoter transcription. We also found evidence that a repressive factor or factors in B cells, but not endothelial or epithelial cells, interacted with more distal elements in the ORF75 promoter region to repress constitutive ORF75 expression in B cells. Alternate forms of Sp1 were found to accumulate during latency and showed substantial enrichment during viral lytic replication in PEL cells and infected endothelial cells, but their functional significance is unclear. We also found that ORF75 can in turn upregulate its own expression and that of other KSHV genes. Thus, while ORF75 acts primarily as a lytic gene in PEL cell lines, Sp proteins induce substantial constitutive ORF75 transcription in infected endothelial cells and this can account for its high expression in KS lesions.

**Data availability statement:** The authors
confirm that all data underlying the findings are
fully available without restriction. All relevant
data are within the paper and its Supporting
Information files.

**Funding:** This work was funded by the
Intramural Program of the NIH, National Cancer
Institute (ZIA BC010885 to RY). The funders
had no role in study design, data collection and
analysis, decision to publish, or preparation of
the manuscript.

**Competing interests:** I have read the journal's
policy and the authors of this manuscript
have the following competing interests: R.
Yarchoan reports receiving research support
from Celgene (now Bristol Myers Squibb),
CTI BioPharma (a Sobi A.B. Company), PDS
Biotech, and Janssen Pharmaceuticals through
Cooperative Research and Development
Agreements (CRADAs) with the National
Cancer Institute (NCI). Dr. Yarchoan also
reports receiving drugs for clinical trials
from Merck, EMD-Serano, and Eli Lilly and
preclinical material from Lentigen Technology
through CRADAs or Material Transfer
Agreements (MTAs) with the NCI. R. Yarchoan
and D.A. Davis are co-inventors on US Patent
10,001,483 entitled "Methods for the treatment
of Kaposi sarcoma or KSHV-induced lympho-
ma using immunomodulatory compounds
and uses of biomarkers." An immediate family
member of R. Yarchoan is a co-inventor on
patents or patent applications related to
internalization of target receptors, epigenetic
analysis, and ephrin tyrosine kinase inhibitors.
All rights, title, and interest to these patents
have been assigned to the U.S. Department of
Health and Human Services; the government
conveys a portion of the royalties it receives
to its employee inventors under the Federal
Technology Transfer Act of 1986 (P.L. 99-502).

## Author summary

In our study, we explored KSHV ORF75, a tegument protein that is increasingly being ap-
preciated as playing a vital role in KSHV replication and inactivating the innate immune
response. ORF75 is characterized as a late lytic protein but has been recently found to be
consistently expressed in Kaposi sarcoma (KS) lesions, which generally express only latent
KSHV genes. Our findings reveal that ORF75 is constitutively expressed in endothelial
and epithelial cells, and to a lesser extent B-cells. We found that constitutive expression of
ORF75 is largely mediated by specificity (Sp) proteins binding to a proximal Sp1 site in
the ORF75 promoter region. Moreover, we found that the lower expression of ORF75 in
KSHV-infected B cells is due to suppressive factors acting at a more distal region of the
ORF75 promoter. We further found that ORF75 can enhance expression of its own RNA
as well as that of several other KSHV proteins, including replication and transcription
activator (RTA) and latency associated nuclear antigen (LANA). This constitutive expres-
sion of ORF75, a late lytic gene, can explain how it can play such an important role in the
earlier steps in KSHV lytic activation and underscores the importance of this gene.

## Introduction

Kaposi sarcoma herpesvirus (KSHV), also called human herpesvirus-8 (HHV8), is the causal
agent for several diseases, including Kaposi sarcoma (KS), primary effusion lymphoma (PEL),
multicentric Castleman disease (MCD), and KSHV-associated inflammatory cytokine syn-
drome (KICS) [1–6]. KS manifests as a vascular tumor characterized by endothelial-like spindle
cells [7]. KS presents in various epidemiologic forms including classical KS (in Mediterranean
regions), endemic KS (in Africa), immune-suppression related KS, and epidemic (HIV-
associated) KS [8–12]. PEL is a subtype of non-Hodgkin lymphoma, characterized by non-solid
tumors within the pleural space and other body cavities. PELs exhibit B-cell features with a
plasmablastic morphology and are primarily observed in individuals in the late stages of HIV/
AIDS [3]. Approximately 80% of PEL tumors are co-infected with Epstein-Barr virus [13].

KSHV is a large double-stranded DNA (dsDNA) gammaherpesvirus virus with an approxi-
mately 140 kb conserved genome. Like other herpesviruses, it has two main gene transcription
programs: latency, in which a few genes are expressed, and lytic replication in which the majority
of genes are expressed, leading to viral replication. The KSHV genome codes for approximately
90 to 100 proteins [14]. In the process of infecting a host cell, KSHV first undergoes limited rep-
lication and then establishes latent phase in which the KSHV DNA exists as a circular chroma-
tinized episome [15,16]. Under certain conditions such as hypoxia, oxidative stress, intracellular
signaling, chromatin modifying agents, or other viral coinfections, viral lytic reactivation can
be triggered [17–20]. During this phase, most viral genes, including immediate early (primary
lytic), early (secondary lytic), and late (tertiary lytic) genes, are expressed in an orderly program,
leading to viral DNA replication, assembly of virion particles, and release of virus.

During lytic reactivation, immediate-early and early lytic genes stimulate the expression
of late lytic genes and also serve to inhibit normal cellular processes and thwart host defense
mechanisms [21–23].Transcription of the late genes of KSHV is a complex process that is
dependent on its DNA replication [24,25]. Late genes encode structural proteins such as
capsid, envelope, and tegument that are essential for the assembly of new virions [24,26].
Late genes typically have minimal promoters, necessitating a virally encoded pre-initiation
complex (vPIC) for efficient transcription [27–30]. While most herpesvirus research has
focused on the orderly program of lytic replication, there is also evidence that some lytic genes

can be directly activated by cellular factors; for example, ORF34 to ORF37 can be activated by hypoxia-inducible factor (HIF), and KSHV-encoded viral IL-6 (vIL-6) and ORF21 can be activated by spliced X-box binding protein-1 (sXBP-1) [31–33].

KSHV ORF75, belongs to the viral formylglycinamide ribonucleotide amidotransferase family and constitutes a crucial component of the KSHV tegument [26,34]. ORF75 is encoded on the antisense strand near the right end of the genome, in close proximity to the latency locus. It exhibits characteristics of and is generally characterized as a tertiary late lytic phase gene, with its RNA and protein levels peaking late in the lytic cycle during lytic reactivation of PEL cells [35–38]. Tegument-associated ORF75 effectively localizes to ND-10 bodies, leading to the degradation of ATRX and dispersion of DAXX, two critical components of the ND-10 host restriction factor complex and thus serves to thwart innate immunity [26]. Moreover, ORF75 along with K13, acts as a co-activator of NF-κB signaling that facilitates KSHV latency [39,40]. Full et al. (2014) have also identified important roles of ORF75 in lytic gene induction, viral genome replication, and virion production. Intriguingly, knocking out or suppressing ORF75 expression in infected iSLK cells, through CRISPR or siRNA techniques, was shown to lead to either complete abolition or significant reduction in viral genome replication, respectively [26,30]. Also, CRISPR knock out of ORF75 yielded a surprising decrease in expression of early as well as late lytic genes [30,41].

While ORF75 has been considered a late-lytic gene, a recent study using bulk RNA sequencing uncovered substantial expression of ORF75 in almost all analyzed KS skin and gastrointestinal lesions [42]. This result is consistent with previous studies also showing ORF75 RNA expression in KS lesions [43–46]. The spindle cells of KS have features of endothelial cells, and RNA-seq analysis of endothelial cells infected with KSHV also revealed a relatively high expression of ORF75 RNA [47]. Intriguingly, BNRF1, the EBV homolog of KSHV ORF75, has similarly been reported to be expressed during latency even though it is generally considered a EBV lytic gene. Also, like studies with KSHV ORF75, BNRF1-KO EBV virus has significantly reduced viral genome replication [48,49]. In a similar manner, ORF75 of MHV68 is expressed in infected macrophages in the absence of other viral gene transcripts [50]. All these examples suggest that there is a complex but poorly understood mechanism of ORF75 expression during latency in certain cell types in multiple gammaherpesviruses.

In this study, we sought to understand the surprisingly high expression of ORF75 in KS lesions, given its previous characterization as a late lytic gene. We show that ORF75 is constitutively expressed in latency but at varying levels in different cell types, with endothelial and epithelial cells showing high expression and B-cells showing lower expression. We further identify multiple Sp1 elements that can affect ORF75 expression and a CAT box within the promoter region of ORF75, with a proximal Sp1 consensus element and two distal CCAAT element, that plays a pivotal role in orchestrating the regulation of ORF75 transcription in various cell types. Finally, we provide evidence that expression of ORF75 during latency plays an important role in the transcription regulation of other KSHV genes.

## Results

### Expression of ORF75 RNA in KSHV-infected cells within KS lesions

Previous studies have shown that KSHV ORF75, a late lytic gene, is consistently expressed in KS lesions, which otherwise have mostly latent gene expression [42,51–53]. To assess whether this was the result of high expression in a few cells or more generalized expression in KSHV-infected spindle cells, we conducted duplex staining of a representative KS-skin lesion section (Fig 1A). ORF75 RNAscope was employed to label ORF75-expressing cells, followed by immunohistochemistry (IHC) of the LANA protein on the same section. Our findings

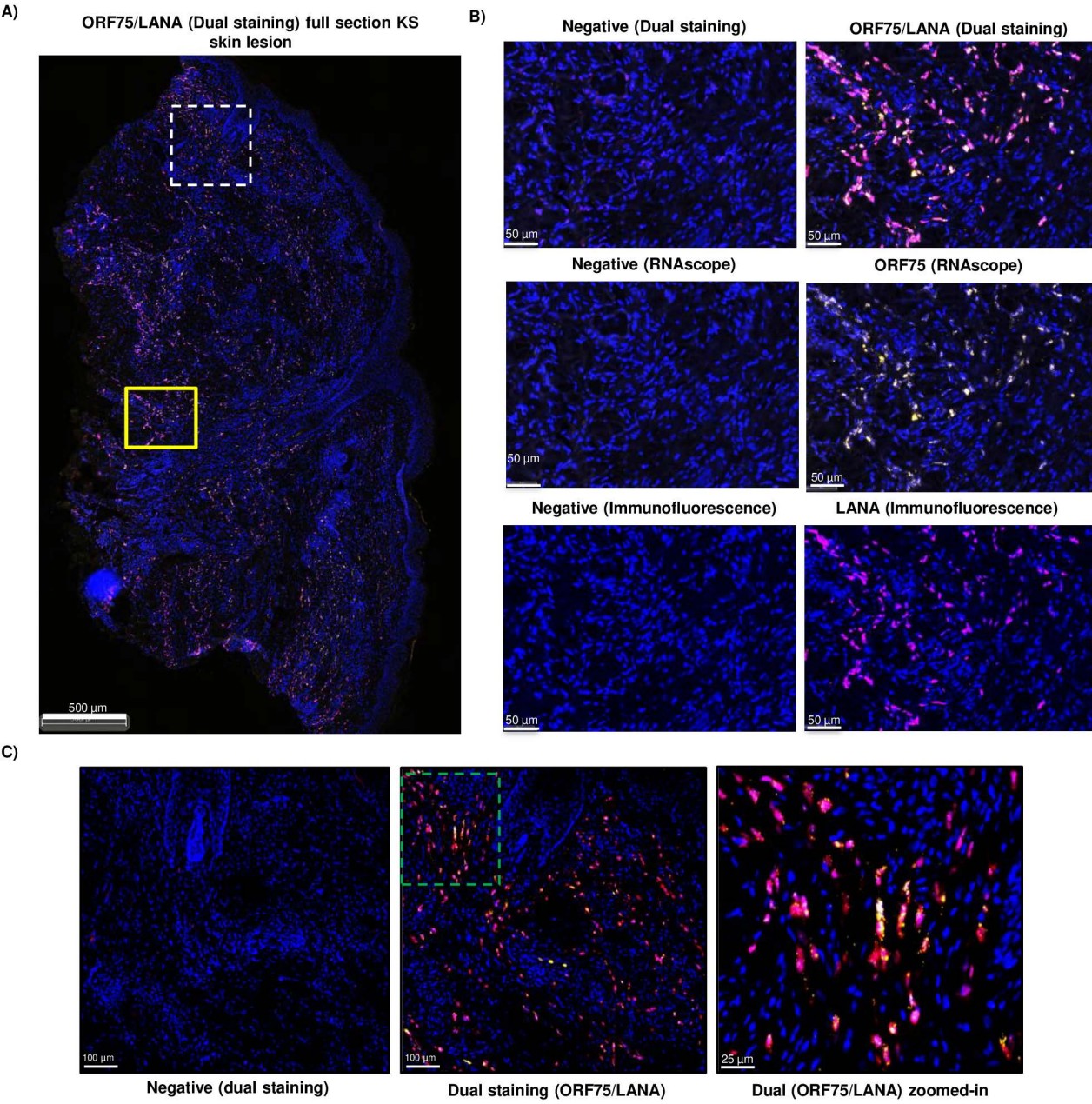

**Fig 1. Presence of ORF75 RNA in nearly all LANA-expressing cells in KS skin tissue. A)** Skin KS lesion from an HIV+ patient with KS and KSHV-MCD, dual-stained with ORF75 RNAscope (yellow) followed by LANA IHC (purple) and counterstained with DAPI (blue). Dual RNAscope-IHC image shown. The yellow box and dashed white box regions are shown in higher magnification in B) and C) respectively. **B)** Higher magnification of yellow box (Fig 1A). Negative staining for RNAscope is with RNAscope 2.5 LS negative control probe *dapB* gene. Negative staining for IHC has no primary antibody. The top panel represents dual RNAscope-IHC image showing negative staining (left) and ORF75 RNA/LANA protein staining (right). The bottom two panels represent same region with individual ORF75 RNAscope and LANA IHC staining, respectively. **C)** The KS-lesion region in the white box is enlarged to show colocalization of ORF75 RNA and LANA protein; the dashed green box is then enlarged further to highlight individual cells.

revealed that the vast majority of infected cells expressing LANA showed colocalization with at least one ORF75 RNA spot (Figs 1B, 1C and S1). We were unable to stain for ORF75 protein due to the lack of a sensitive ORF75 antibody for use in IHC.

## ORF75 RNA is expressed at high levels in latently infected endothelial cells compared to primary effusion lymphoma cell lines

KS tumors are characterized by spindle-like cells expressing endothelial markers, whereas PEL cells are derived from B-cells. We hypothesized that the ORF75 expression in KS lesions might be regulated by cell-specific factors. To investigate this, we established a latently infected endothelial cell line (TIME.219) by infecting telomerase-immortalized microvascular endothelial (TIME) cells with recombinant KSHV.219 virus [54] constitutively expressing GFP from an EF1α promoter and RFP from the lytic PAN promoter (Fig 2A). Through qPCR-based expression analysis, we compared expression levels of representative genes from each category of lytic genes (primary, secondary, and tertiary) in TIME.219 cells with expression of the genes in BCBL-1, a PEL cell line. Consistent with the finding in KS lesions, ORF75 expression was notably higher with relation to LANA in the latently infected endothelial cell line (TIME.219) as compared to other lytic genes. (Fig 2B). Conversely, BCBL-1 cells exhibited relatively lower ORF75 expression, approximately at the same level as LANA (Fig 2B). We also examined the iSLK-BAC16 line, now recognized to be of epithelial origin, and found that it also displayed some ORF75 expression, higher than the BC3 PEL cell line but lower than the expression in TIME.219 line (S2A Fig). The ORF75 gene is transcribed both as part of the bicistronic K15 RNA and independently via its own promoter. We assessed K15 expression by analyzing RNA levels from its exons 1 and 3 and compared them to ORF75 RNA across various infected cell types. ORF75 RNA consistently showed at least two-fold higher expression than K15 RNA, supporting ORF75 gene's independent regulation via its own promoter (S2B Fig).

During primary infection, KSHV generally has a period of lytic gene expression and then establishes latency within 3-5 days, depending upon culture conditions. To examine the expression of ORF75 during primary infection, we performed *de novo* infection of TIME cells with rKSHV.219 virus and analyzed representative genes at 1-, 3-, and 6-days post-infection. As expected, most viral lytic genes peaked during the lytic phase of infection and were then more quiescent as latency was established. By contrast, ORF75 expression gradually rose and was highest at day 6, similar to ORF72, a latent gene (Fig 2C). Similarly, we *de novo* infected primary human dermal microvascular endothelial cells (HDMVEC) with rKSHV.219 and observed an ORF75 expression pattern consistent with that in *de novo* infected TIME cells (S2C Fig). These results suggest that ORF75 gene expression is regulated independently of other lytic genes or of viral DNA replication and is expressed without the requirement of a vPIC in endothelial cells.

## The ORF75 promoter has high uninduced basal transcription activity

We hypothesized that a cellular transcription factor is responsible for inducing the expression of the ORF75 gene during latency. We analyzed the ORF75 promoter for various transcription factor binding elements using Geneious Prime software's TF scan function against a custom transcription factor database using EMBOSS nucleotide package (Fig 3A). We identified a number of sequences for transcription response elements including hypoxia response elements (HREs), antioxidant response elements (AREs), and STAT-like elements among others, potentially capable of driving the ORF75 promoter independently of the vPIC complex. To investigate these possibilities, we first created promoter luciferase fusion constructs with ~1.2 kb ORF75 promoter fused in both forward and reverse orientations with a luciferase reporter gene (Fig 3B). These constructs were transiently transfected into 293T cells, followed by a luciferase assay for quantifying promoter activity. Surprisingly, ORF75's promoter exhibited high uninduced basal transcription activity in uninfected epithelial cells. This basal activity was only observed with the forward fusion promoter construct, but not

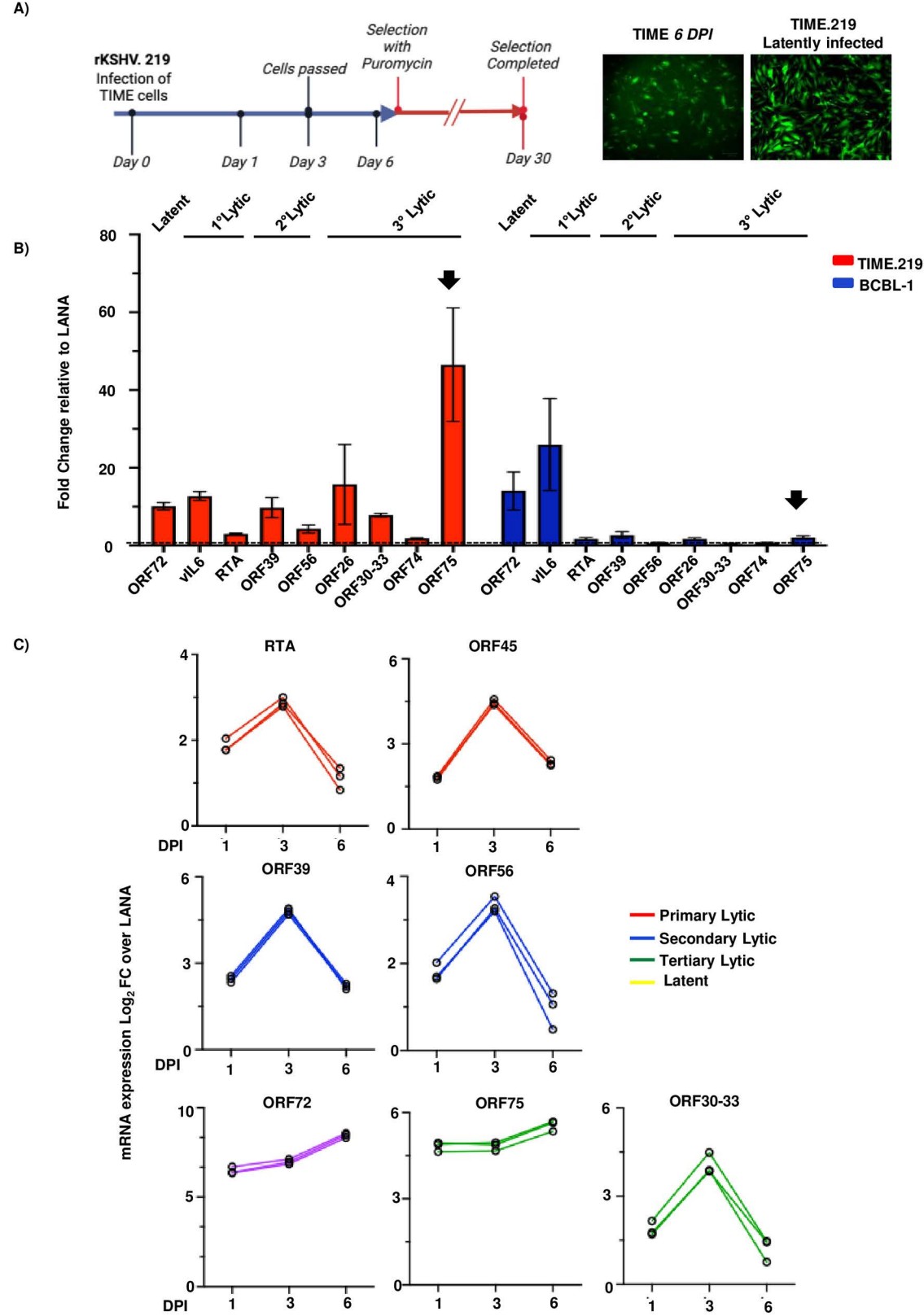

**Fig 2. ORF75 RNA is constitutively expressed in latently infected endothelial cells. A)** Schematic of the establishment of the TIME.219 KSHV-infected immortalized endothelial cell line by infection of TIME cells with recombinant KSHV.219 virus. Blue

and red arrows indicate the period of infection without and with puromycin selection, respectively. Sample collection days during the initial period of selection are shown. Representative images of *de novo* infection at day 6 and after stable selection (TIME.219) are shown. **B)** qPCR analysis of representative KSHV latent and lytic genes in the latently infected TIME.219 cell line and the BCBL-1 (PEL) cell line. N=3 biological replicates of the established cell lines TIME.219 (obtained after day 30) and BCBL-1. qPCR analysis was done with 40 cycles and a GAPDH internal reference control. Expression was normalized to corresponding LANA expression of TIME.219 and BCBL-1, respectively. Error bar indicate ± standard deviations of 3 separate experiments. Black arrows highlight the ORF75 gene. **C)** qPCR analysis of representative latent and lytic KSHV genes at 1, 3, and 6 days post infection (DPI) after *de novo* infection of TIME cells with rKSHV.219 virus. Connected lines indicate gene expression trend of the same gene throughout the course of experiment.

the reverse promoter construct (Fig 3C), suggesting that the element responsible for basal activity might be either unidirectional or located toward the proximal region of the promoter. We next compared ORF75's promoter activity with RTA (ORF50), ORF57 (immediate early), LANA, ORF72 (latent) and ORF74 (late) promoters (Figs 3D and S3A). As previously reported [55,56], RTA and LANA promoter showed high basal activity in 293T cells; however, the ORF75 promoter showed the highest constitutive activity overall. The ORF74 promoter showed negligible basal expression while the ORF72 and ORF57 promoters showed relatively low basal activity. We then compared the basal activity of the ORF75 promoter with a promoter of the same length of another lytic gene, ORF74 in two different epithelial cell lines. ORF75's promoter demonstrated significantly higher basal promoter activity in both the tested cell lines compared to the negligible promoter activity of ORF74, which was similar to the control vector without any promoter (S3B Fig).

We further examined the basal expression mediated by the ORF75 promoter in HepG2, COS-7, and SLK epithelial cell lines, and found relatively high basal activity in these lines as well (S3C Fig). We hypothesized that the high basal activity of the ORF75 promoter in these cell lines may be attributed to the presence of HRE, ARE, and STAT elements within the promoter region. We performed multiple redundant experiments to test any effect of the HRE and ARE elements on the ORF75's promoter activity. HRE elements were tested by HIF (1 & 2) over-expression and under hypoxia, ARE elements were tested by NRF2 over-expression, $H_2O_2$, tertiary butylhydroquinone (tBHQ) and beta-naphthoflavone (β-NF). These conditions showed little or no enhancement of the already high basal activity (S3D, S3E, S3F, S3G Fig). To conclude any role of these elements in ORF75's basal promoter activity, we mutated these elements within the full-length ORF75 promoter and assessed the changes in the basal transcription levels. We did not observe a significant change in the basal transcription activity of the ORF75 promoter upon mutating individual ARE elements or all ARE, HRE, and STAT-like elements together (Fig 3E). These results suggest that the late gene ORF75 promoter is highly active in cells lacking other viral transactivators, and that this basal activity is not attributed to the HRE, ARE, or STAT elements within the ORF75 promoter.

## A proximal Sp1 element is essential for basal transcription activity of the ORF75's promoter

We next performed a series of promoter truncation analysis in a bid to identify the minimal region required for the basal promoter activity of ORF75. Truncations p75-T1 through p75-T4 showed some loss but p75-T4, which contained only the proximal 200 bp, still had substantial basal activity in 293T cells (Fig 4A). By contrast, promoter truncation p75-T5, which lacked the proximal 200 bp region but included the region from -49 to -269, showed a complete loss of basal activity. Taken together, these results provided evidence that the proximal 200 bp region of the ORF75 promoter is largely responsible for the basal transcription activity of the promoter in epithelial cell lines. Examination of the proximal 200 bp region identified

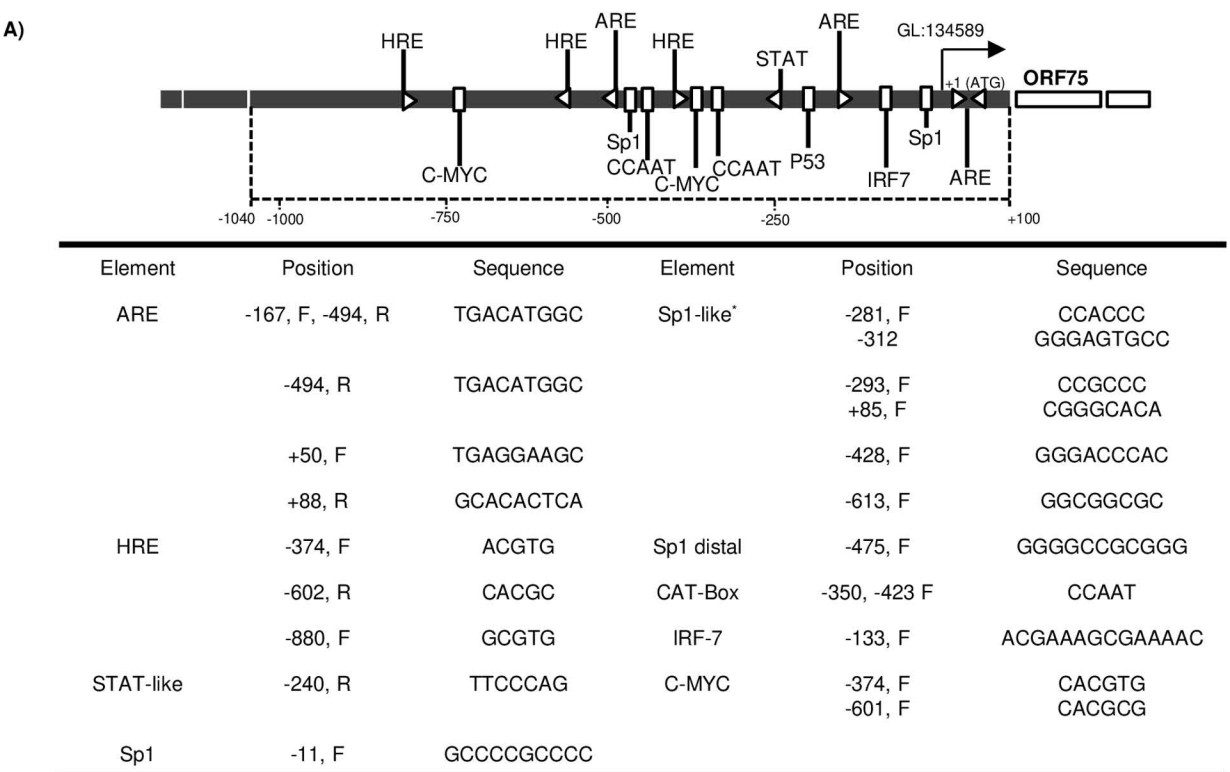

| Element | Position | Sequence | Element | Position | Sequence |
|---|---|---|---|---|---|
| ARE | -167, F, -494, R | TGACATGGC | Sp1-like* | -281, F -312 | CCACCC GGGAGTGCC |
| | -494, R | TGACATGGC | | -293, F +85, F | CCGCCC CGGGCACA |
| | +50, F | TGAGGAAGC | | -428, F | GGGACCCAC |
| | +88, R | GCACACTCA | | -613, F | GGCGGCGC |
| HRE | -374, F | ACGTG | Sp1 distal | -475, F | GGGGCCGCGGG |
| | -602, R | CACGC | CAT-Box | -350, -423 F | CCAAT |
| | -880, F | GCGTG | IRF-7 | -133, F | ACGAAAGCGAAAAC |
| STAT-like | -240, R | TTCCCAG | C-MYC | -374, F -601, F | CACGTG CACGCG |
| Sp1 | -11, F | GCCCCGCCCC | | | |

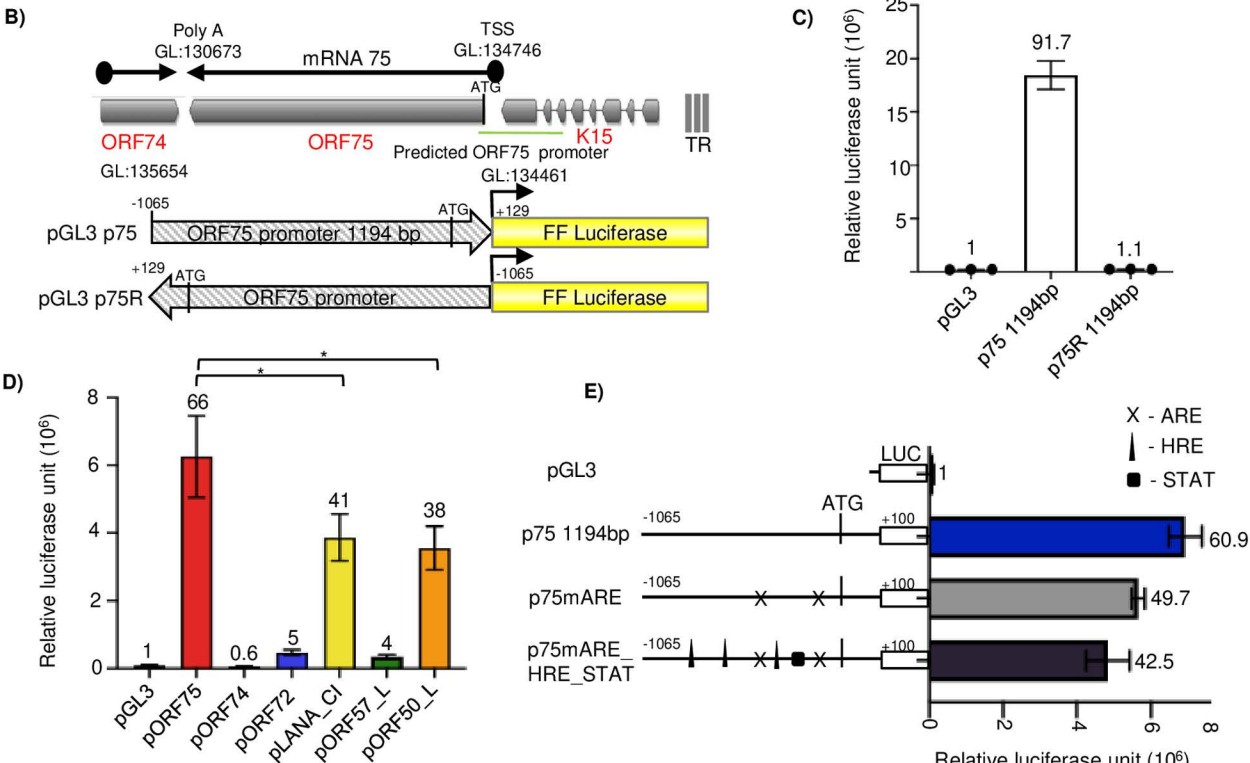

**Fig 3. ORF75 promoter has a very high basal transcription activity. A)** (Top) Schematic of the ORF75 promoter region with various putative transcription factor binding elements. (Bottom) Table showing the genomic location, orientation and sequence of various promoter elements with respect to ORF75's start codon (ATG). **B)** Schematic showing a map of the ORF75 protein coding transcript and the ORF75 promoter-luciferase

plasmids. Note that the gene map is shown in the conventional orientation, but the plasmids are shown in the opposite orientation. **C)** Promoter luciferase assay of ORF75 full-length (p75) construct in unstimulated HEK293T cells. pGL3 is the empty vector control. Assayed at 4 days post transfection. **D)** Promoter luciferase assay of ORF75, ORF74, ORF72, LANA, ORF57 and RTA promoter activity in uninfected HEK293T cells. Assayed at 72h post transfection. **E)** Promoter luciferase assay in HEK293T cells of various ORF75 promoter mutation constructs. Schematics of the mutation constructs are shown on the left and the corresponding basal transcription activity on the right. Assayed at 72h post transfection. Numbers on the top of each bar in Fig 3C, 3D and 3E indicates average fold change normalized to pGL3 vector control. Shown are the means ± standard deviations of at least 3 separate experiments. *P*-values (*$p \leq 0.05$, ns not significant) are calculated using two-sided paired *t*-test. Relative luciferase unit indicated is normalized to Beta-galactosidase that was used as transfection control. GL: genomic location as per NC_009333 KSHV reference genome.

three elements of interest: a putative Sp1 element and two ARE motifs, the latter each with one mismatch from the ARE consensus sequence. Two mutations of p75-T4 were made, one deleting the proximal Sp1 (pSp1) element and the other mutating the two ARE elements. Promoter p75-T4δSp1, lacking the pSp1 element, lost nearly all of the basal promoter activity. By contrast, the promoter truncation with the intact Sp1 element but with mutated ARE motifs still had activity similar to p75-T4 (Fig 4A and 4B). These results suggest that the pSp1 element, located at positions -11 to -21 bp relative to the ORF75 ATG codon, is essential for the ORF75 promoter's basal transcription activity.

### Sp transcription factors directly bind to and regulate the Sp1 element of the ORF75's promoter

The Sp1 element, primarily regulated by the Sp family of proteins, functions as a GC-box. Among the Sp protein family members, Sp1, Sp3, and Sp4 are prominently expressed and regulate target genes in a tissue-specific manner by binding to the Sp1 element. To ascertain whether one or more Sp proteins, in fact, bind to the pSp1 element of the ORF75 promoter, we employed an electrophoretic mobility shift assay (EMSA). A 50 nt dsDNA probe labeled with infrared 680 dye, corresponding to the proximal region of the ORF75 promoter containing the pSp1 element (Fig 4C), was incubated with lysates overexpressing Sp proteins. The binding pattern of the Sp proteins from the 293T cell lysates was determined by using a commercial probe containing an Sp1 element (S4A and S4B Fig). We then found that Sp1, Sp3, and Sp4 proteins demonstrated direct binding to the p75 pSp1 ORF75 promoter probe (Fig 4D). To verify the specificity of binding, we utilized an excess of unlabeled (cold) probe containing either the wild-type pSp1 or a mutated pSp1 element. As anticipated, binding of Sp proteins to the ORF75 promoter probe was completely abolished upon titration with the wild-type pSp1 site-containing cold probe, whereas binding persisted with the mutated pSp1 site-containing probe (Fig 4E). Moreover, none of the Sp proteins exhibited binding to the ORF75 promoter probe with a mutated pSp1 element, indicating the specificity of Sp protein binding to the ORF75 promoter pSp1 element (Fig 4E).

As described above, we observed expression of ORF75 RNA in latently infected iSLK-BAC16 cells, and also basal transcription activity of the ORF75 promoter in iSLK cells. Sp1 and Sp3 can interact with and recruit transcriptional complexes in Sp1 element regulation [57]. To investigate whether this occurs in the ORF75 promoter during KSHV latency, we conducted a chromatin immunoprecipitation assay (ChIP) using Sp1 and Sp3 antibodies, alongside a histone H3 antibody control, on uninduced iSLK-BAC16 cells. We assessed ORF75 genic region enrichment via qPCR. As a positive control, we analyzed the cellular DHFR promoter, previously known to be regulated by Sp1 and Sp3 proteins. As anticipated, the DHFR promoter exhibited significant enrichment of both Sp1 and Sp3 proteins. Similarly,

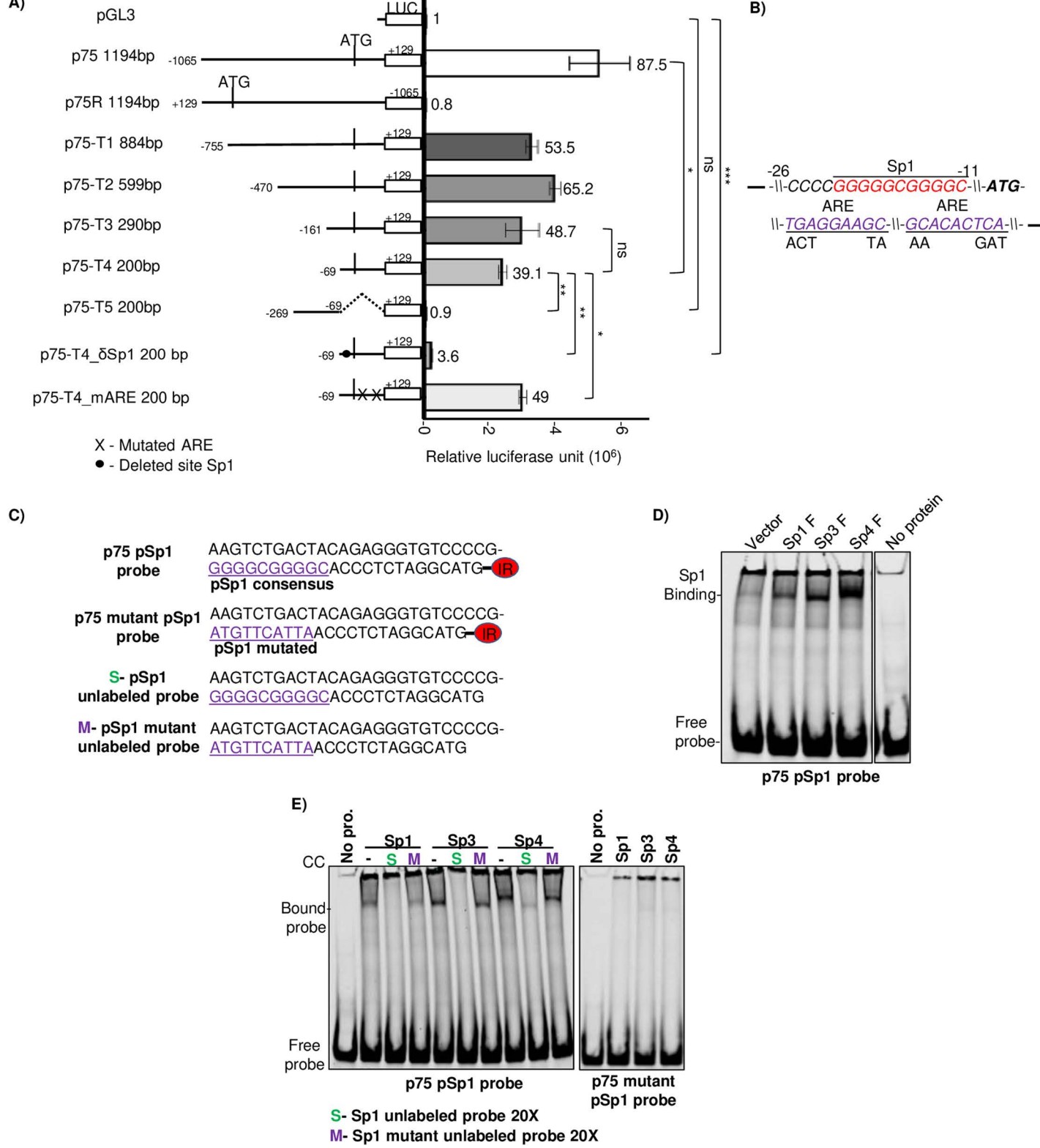

**Fig 4. ORF75 promoter's high basal transcription activity is regulated by Sp1 DNA elements. A).** (Left) schematic of truncation mutants, deletion mutants, and binding site mutants of the ORF75 promoter used to make luciferase constructs. (Right) promoter luciferase activity of these constructs in 293T cells. Numbers on the top of each bar indicates average fold change relative to empty vector pGL3 control. **B)** Schematic showing the location and sequence of the proximal Sp1 (pSp1) element and ARE elements in the ORF75 promoter and the mutants. **C)** Sequences of the pSp1 consensus element of ORF75 promoter probe and the mutant probes

used in EMSA. The promoter probe is a 50 nt long dsDNA labelled with 5' IR dye CW700. The mutated pSp1 element is in purple and underlined. The terminal ATG is the start codon of ORF75 protein. **D)** EMSA showing binding of Sp1, Sp3, and Sp4 proteins with the double-stranded DNA (dsDNA) IR-probe of ORF75 promoter in a 8% native PAGE gel. Equal concentration of HEK293T lysates over-expressing the various Sp proteins were used in the assay. **E)** Same as D), but with competitive specific and Sp1 element mutated probes. Shown are the means ± standard deviations of at least 3 separate experiments. *P*-values (*$p \leq 0.05$, **$p \leq 0.01$, ns not significant) are calculated using two-sided paired *t*-test. Representative gel shift assay blots shown. All gel shift assay observations were reproduced in at least 3 separate experiments. Beta-galactosidase was used for transfection normalization. See S7 Fig for full blots.

both Sp1 and Sp3 proteins were also enriched on the ORF75 promoter, although not to as great an extent (S4C and S4D Fig).

Next, we examined whether overexpression of Sp proteins would affect the basal promoter activity of ORF75. We over-expressed Sp1, Sp3, and Sp4 proteins along with ORF75 promoters of two different lengths. All Sp proteins were successfully expressed although Sp1 showed relatively less overexpression (Fig 5A). Interestingly, overexpression of each of these Sp proteins Sp1, Sp3, and Sp4 further induced the ORF75 promoter. Sp3 preferentially enhanced the shorter p75-T2 promoter over the full-length p75, suggesting differential Sp3 binding sites in the ORF75 promoter with activating and repressive functions. Sp4 induced particularly strong enhancement of the ORF75 promoter (Fig 5A). Sp1, known to form homotetramers, synergizes with other Sp proteins and transcription factors to hyperactivate promoters under various conditions. We hypothesized that Sp1 might be the primary Sp protein regulating ORF75 promoter activity in endothelial cells. To test this, we employed mithramycin A (MA), a specific inhibitor of Sp1 protein that reduces Sp1 protein levels and interferes with its binding to GC-box [58]. Treatment with MA significantly decreased ORF75 promoter activity by over 70%, supporting Sp1's role as a key regulator of ORF75 promoter activity (Fig 5B). We further performed supershift assays for pSp1 element with different Sp1 and Sp3 antibodies. The Proteintech (AP) Sp1, but not the Sp3 antibody, inhibited the formation of the protein-DNA complex (Fig 5C). As a 50nt long dsDNA is used for these assays, the antibody complex formed can be of very high molecular weight and might sit in the well. To verify our supershift observation, we used a few other Sp1 antibodies (1C6 and E3, Santa Cruz) and longer run periods. The supershifted Sp1 complex was distinctly observed under these conditions (Fig 5D). These findings suggest that Sp1 transcription factor likely serves as a key inducer of the ORF75 promoter's basal activity, at least in HEK293T cells.

## Sp1 transcription factor is essential for the basal transcription activity of the ORF75 promoter

We hypothesized, based on our observations, that there should be little or no basal ORF75 promoter activity in cells lacking Sp1 protein. To test this hypothesis, we utilized the *Schneider Drosophila cell line 2* (SL2), which is deficient in Sp family proteins or homologs capable of regulating Sp1 elements [59,60]. Utilizing the ORF75 promoter luciferase assay, we observed no basal transcription activity of the ORF75 promoter in SL2 cells (Fig 6A). However, when we co-expressed human Sp1 protein, basal transcription activity was restored. To confirm that the effect of exogenous Sp1-TF protein occurs via the pSp1 element of the ORF75 promoter in the SL2 cell line, we used an ORF75 promoter with the pSp1 site deleted (p75-T4δSp1) and observed no significant promoter activity even in the presence of exogenous Sp1 protein (Fig 6B).

As we have seen, truncations of the ORF75 promoter exhibit varying basal transcription activity, even though all truncations, except p75-T5, contain the proximal Sp1 element. This

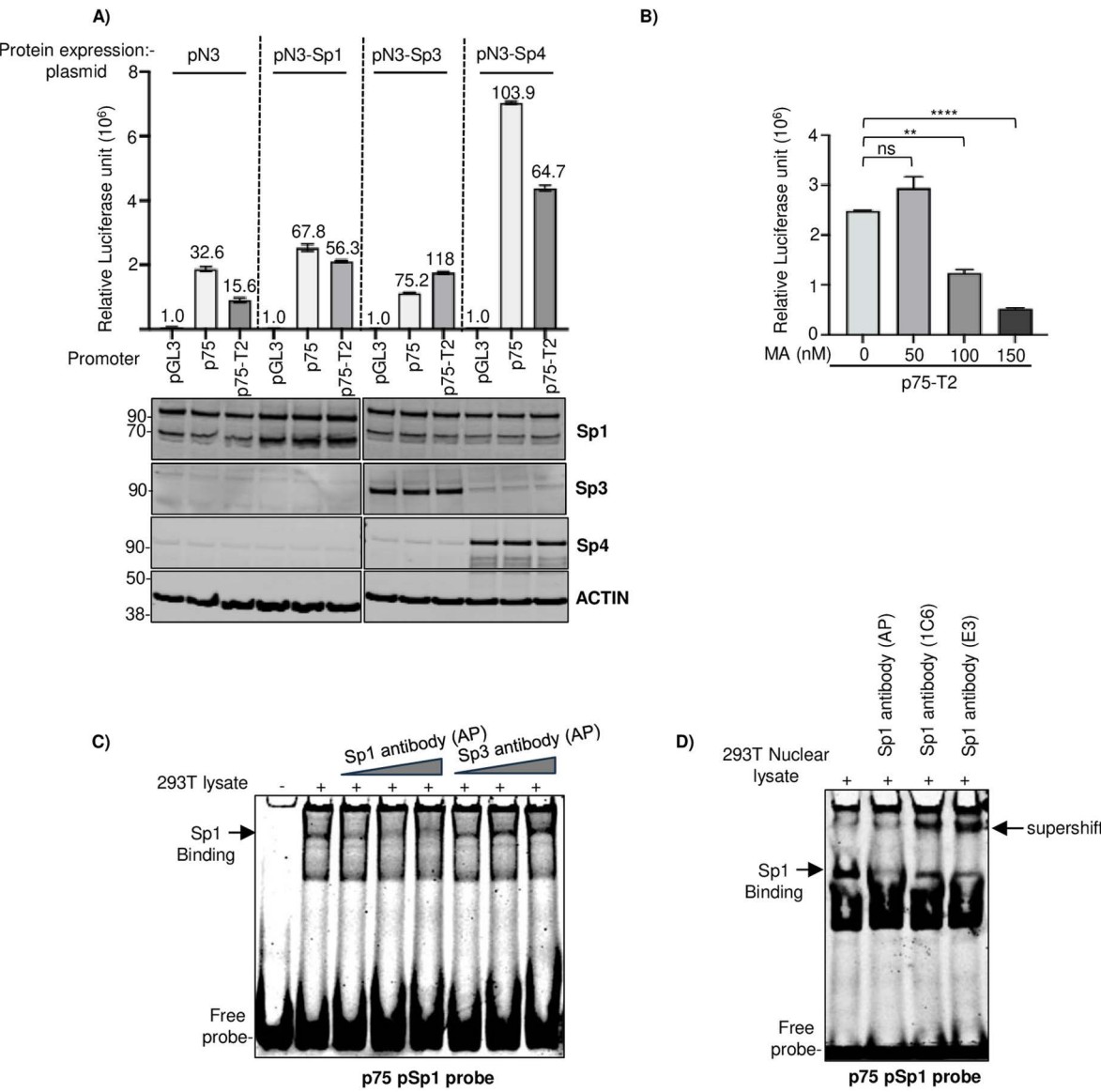

**Fig 5. ORF75 promoter's Sp1 DNA elements are regulated by Sp transcription factors.** **A)** Promoter luciferase assay of ORF75 promoter constructs along with over-expression of various Sp proteins in HepG2 cells. Sp proteins were expressed from a CMV-driven pN3 vector. ORF75 full-length (p75) and truncated (p75-T2) promoters are shown. Numbers on the top of each bar represents average fold change normalized to untreated pGL3 for each group of expression plasmid set as 1. The panel below the histogram represents the representative western blot for the over-expression proteins. **B)** ORF75 promoter (p75-T2) luciferase activity in the presence of Sp1-specific inhibitor Mithramycin A in 293T cells. 48h post transfection. **C)** Supershift assay. Equal amounts of HEK293T lysates (30ug) were preincubated with 1, 1.5 and 2µg of anti-Sp1 (21962-1-AP, Proteintech) or anti-Sp3 (26584-1-AP, Proteintech) before incubation with ORF75 promoter probe (p75 pSp1 probe) followed by gel shift assay. 6% native TBE gel. **D)** Same as in C) but with HEK293T nuclear lysates and 3µg each of different Sp1 antibodies. Sp1 antibody (AP) (21962-1-AP, Proteintech), Sp1 antibody (1C6) (sc-420, Santa Cruz), Sp1 antibody (E3) (sc-17824, Santa Cruz). Shown are the means ± standard deviations of 3 separate experiments. *P*-values (**$p \leq 0.01$, ****$p \leq 0.0001$, ns not significant) are calculated using two-sided paired *t*-test. Beta-galactosidase was used for transfection normalization. See S7 Fig for full blots.

was observed in both the 293T cell line (Fig 4A) and the SL2 cell line (Fig 6B) when we utilized exogenous human Sp1, suggesting that other sites may also be affected by the Sp1 transcription factor. To map the activity of other Sp1 sites, we mutated a putative distal full-length Sp1 element (dSp1), along with proximal consensus Sp1 (pSp1) site, multiple Sp1-like sites, and

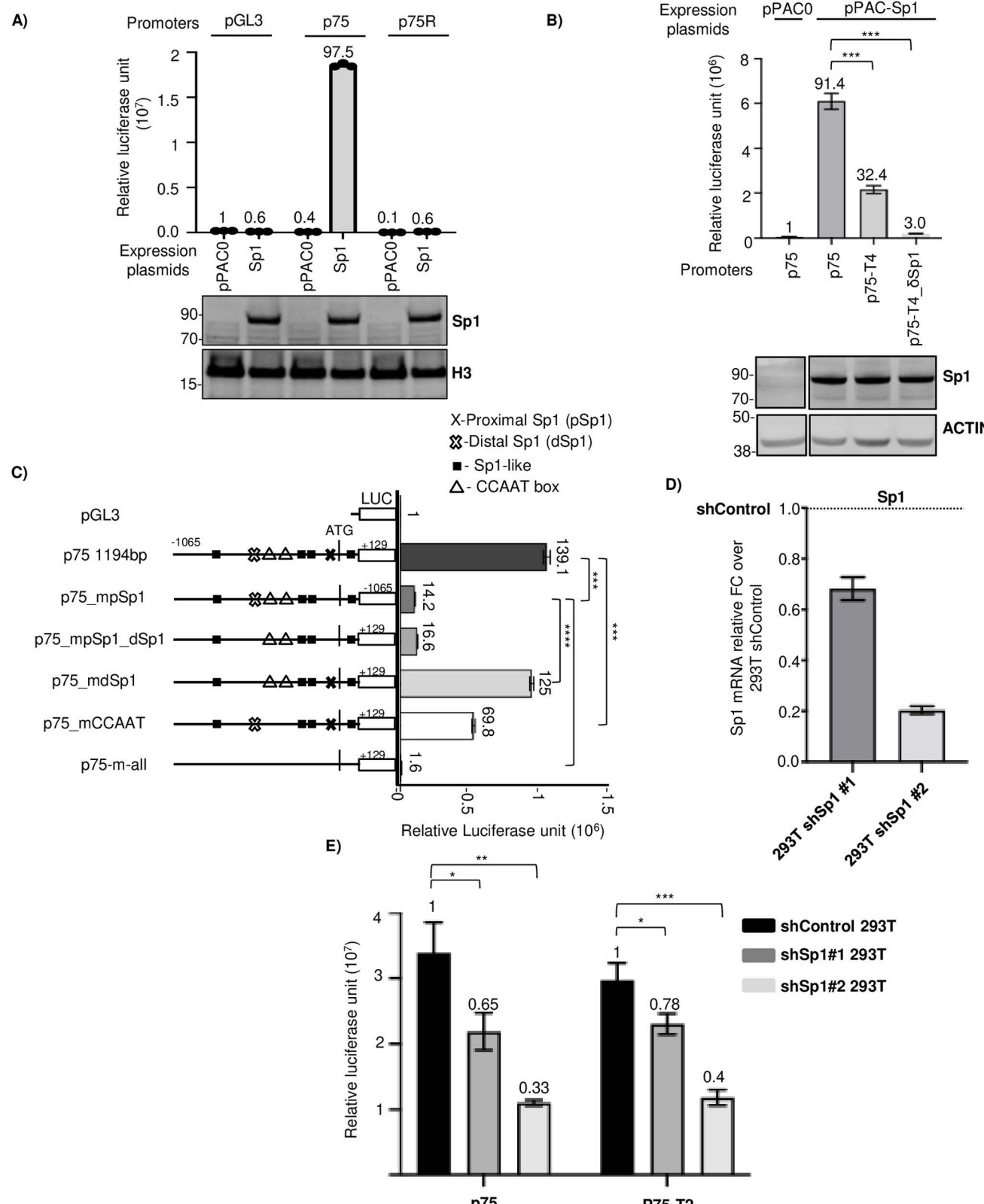

**Fig 6. Sp1 protein is the key transcription factor regulating ORF75 promoter's basal activity. A)** Promoter luciferase assay of ORF75 promoter in *Schneider Drosophila line 2* (SL2) with or without exogenous human Sp1 expression. Sp1 protein was expressed from pPAC vector driven by ACTIN 5C promoter. pPAC0 is the empty vector. Numbers on the top of each bar represents average fold change normalized to pGL3 with pPAC0 set as 1. Assayed

at 72h post nucleofection. The western blot panel below shows the expression of exogenous Sp1 protein in SL2 cells. Data are presented as mean ± SD of three individually nucleofected samples. **B)** Same as in A), except with ORF75 p75-T4 truncated and Sp1 element-deleted promoter constructs. Numbers on the top of each bar represents average fold change normalized to p75 with pPAC0 set as 1. **C)** Promoter luciferase assay of various mutant ORF75 promoters in *Schneider Drosophila line 2* (SL2) with exogenous human Sp1 expression. Presence of various elements are shown in the promoter schematic as different shapes. Absence of a shape in the promoter schematic indicates corresponding element mutation. Data shown are presented as mean ± SD of three individually nucleofected samples. Assayed at 72h post nucleofection. Numbers on the top of each bar represents average fold change normalized to pGL3 set as 1. *P*-values (***$p \leq 0.001$, ****$p \leq 0.0001$, ns not significant) are calculated using two-sided paired *t*-test. **D)** qPCR expression analysis of Sp1 mRNA in shControl 293T and two shSp1 293T stable Sp1 knockdown cell lines. Sp1 mRNA expression is normalised to shControl. Internal reference gene GAPDH. **E)** Promoter luciferase assay of full length ORF75 promoter (p75) and truncated promoter (p75-T4) in shControl and shSp1 293T knockdown cell lines. Assayed at 3 DPI. Shown are the means ± standard deviations of 3 separate experiments. Numbers on the top of each bar represents average fold change normalized to shControl values for p75 and p75-T2 set as 1. *P*-values (*$p \leq 0.05$, **$p \leq 0.01$, ***$p \leq 0.001$, ns not significant) are calculated using two-sided unpaired *t*-test. Beta-galactosidase was used for transfection normalization. See S7 Fig for full blots.

two CAT boxes (Fig 6C, schematics). The SL2 cell line with exogenous human Sp1 protein was used to directly assess the response of various ORF75 promoter's Sp1 DNA elements to Sp1 protein. In the presence of a strong pSp1 element, distal CAT boxes can enhance Sp1-mediated transcription activity. Mutating the pSp1 element in the full-length promoter abolished 90% of basal transcription activity (Fig 6C). By contrast, mutating the dSp1 element had little or no effect on basal transcription in the SL2 cell line, suggesting that Sp1 protein does not substantially regulate this site by itself. Mutating the two CAT boxes alone in the full-length promoter (p75_mCAAT) resulted in about a 50% reduction in promoter activity. Mutating the pSp1, the dSp1, all Sp1-like sites, and two CAT boxes completely abolished ORF75's Sp1-dependent promoter activity. These results indicate that the binding of Sp1 protein to the pSp1 site is the most critical step for ORF75 promoter activity, that the CAT boxes further enhance Sp1-mediated promoter activity only in the presence of the proximal Sp1 element, and that the dSp1 site has little or no effect on ORF75 promoter activity. This suggests that Sp1 can coordinate multiple regions of ORF75 promoter to regulate its expression. A limitation of the SL2 system is that we were only able to measure the activation and cooperativity of different Sp1 binding sites, but not repression, due to the lack of a repressor that can coordinate with exogenous human Sp1 protein. It thus remained possible that a repressor could coordinate with Sp1 to regulate and repress the CAT boxes or the dSp1 site.

While experiments in the SL2 cell line clearly substantiate that Sp1 activates the ORF75 promoter, they do not exclude the involvement of other GC-box binding factors absent in SL2 cells. To further investigate, we generated stable Sp1 knockdown 293T cell lines using lentiviral shRNA targeting Sp1. Since Sp1 is essential for cell growth and homeostasis, complete knockout can be lethal. We isolated two shRNA Sp1 colonies (shSp1) and a control shRNA (non-targeting) colony (shControl). shRNA Sp1#1 showed 35% knockdown efficiency, while shRNA Sp1#2 achieved 80% knockdown compared to the control (Fig 6D). Promoter activity of the full-length ORF75 promoter (p75) and the truncated version (p75-T2) was significantly reduced in Sp1 knockdown cells. In the shRNA Sp1#2 line with 80% Sp1 knockdown, promoter activity was reduced by 67% (Fig 6E). These results indicate that Sp1 is a key activator required for the basal activity of the ORF75 promoter.

### The ORF75 promoter has significantly higher basal transcription activity in endothelial cells than B-cells

The latently infected endothelial cell line (TIME.219) exhibits relatively high expression of the ORF75 gene compared to the PEL cell lines (BCBL-1 and BC3). To look at differences in the ability of different cell types to activate ORF75, we investigated the ORF75 promoter activity in endothelial and B-cell lines using a promoter luciferase assay. ORF75's basal

transcription activity was significantly higher in TIVE (telomerase-immortalized vascular endothelial cells) [61] compared to BJAB B-cells (Fig 7A). We hypothesized that this difference in basal activity might be attributed to varying levels of Sp proteins in these cell types and that endothelial cells would have a higher level of Sp1 than B cell lines. Surprisingly, there were higher Sp1 levels in each of 3 B-cell lines tested compared to endothelial cell lines TIME and TIVE (Fig 7B) or with primary endothelial cells (HUVEC) (S4E Fig). Interestingly, we noted a drastic difference in Sp4 protein levels among the uninfected BJAB cell line and the KSHV-infected BCBL-1 and BC3 cell lines. Even with these results, we questioned whether Sp1 protein activity might differ across these lineages, as active Sp1 protein is localized in the nucleus, and looking at total cellular Sp1 protein might not accurately represent Sp1 protein activity. Active Sp1 protein is the portion of Sp1 that is localized in the nucleus and is able to modulate transcription by binding to the GC-box or other Sp1 elements in the DNA. To assess active Sp1 protein in various lineages of uninfected and latently infected cell lines, we utilized nuclear lysates and performed EMSA using an ORF75 pSp1 element probe, while simultaneously examining nuclear Sp1 protein levels by western blot (WB) (Fig 7C). Sp1 binding to the pSp1 element of the ORF75 promoter was strongest in the epithelial cell lines iSLK and 293, followed by the B-cell lines BJAB and BCBL-1. Surprisingly, it was lowest in the endothelial cell line (TIME). We also observed somewhat higher binding in latently infected lines compared to their uninfected counterparts (Fig 7D). Overall, these results suggested that there was poor correlation between Sp1 binding to the proximal promoter and varying ORF75 promoter activity in various cell lines.

## The ORF75 promoter is repressed in B-cells compared to endothelial cells

Based on the above experiments, neither differences in Sp1 protein levels nor binding of active Sp1 protein appeared to explain the higher activity of the ORF75 promoter in endothelial cells compared to B cells. We wondered whether these differences could be due to differential repression of the promoter. To explore this, we performed a promoter-luciferase assay in BJAB and TIVE cell lines, testing the basal promoter activity of the full-length ORF75 promoter construct and comparing it with the p75-T4 truncated promoter construct that contains only the initial 200 bp region with the pSp1 element but lacking the distal elements (Fig 7E). As expected, the full-length ORF75 promoter showed substantially (about 3.6-fold) higher activity in the endothelial cell line (TIVE) than in the B-cell line (BJAB). Interestingly, the basal activity of the truncated promoter p75-T4 was about 1.5-fold higher than that of the full-length promoter in TIVE cells, suggesting that the distal region of the ORF75 promoter is under some repression. In BJAB B-cells, the p75-T4 truncated promoter showed almost 4-fold higher activity than the full-length promoter, suggesting strong repressive effect of the distal promoter region in those cells. Deleting the pSp1 element in the truncated p75-T4 promoter (p75-T4_ΔSp1 200 bp) abolished substantial activity in both cell lines. Further, mutating pSp1 element in the full-length promoter (p75-mSp11194 bp), resulted in a complete loss of basal activity in the B-cell line but only reduced about half the basal activity compared to the full-length promoter in the endothelial cell line (TIVE). These results suggest that the ORF75 full length promoter is repressed by an element in its distal region and that this repression is substantially higher in B-cells. It also suggested that there is some ORF75 promoter activity in TIVE cells that is mediated by a promoter area distal to the pSp1 element, and that this activity mediated by the distal area is not present in BJAB cells.

In an effort to identify the transcriptional repressor of ORF75 promoter, we tested multiple potential repressive transcription factors with consensus DNA elements in the ORF75 promoter or that have been shown to repress Sp1 activity in other systems. Interestingly, BCL6, a B-cell specific protein, showed substantial repression of the ORF75 promoter (S5A Fig).

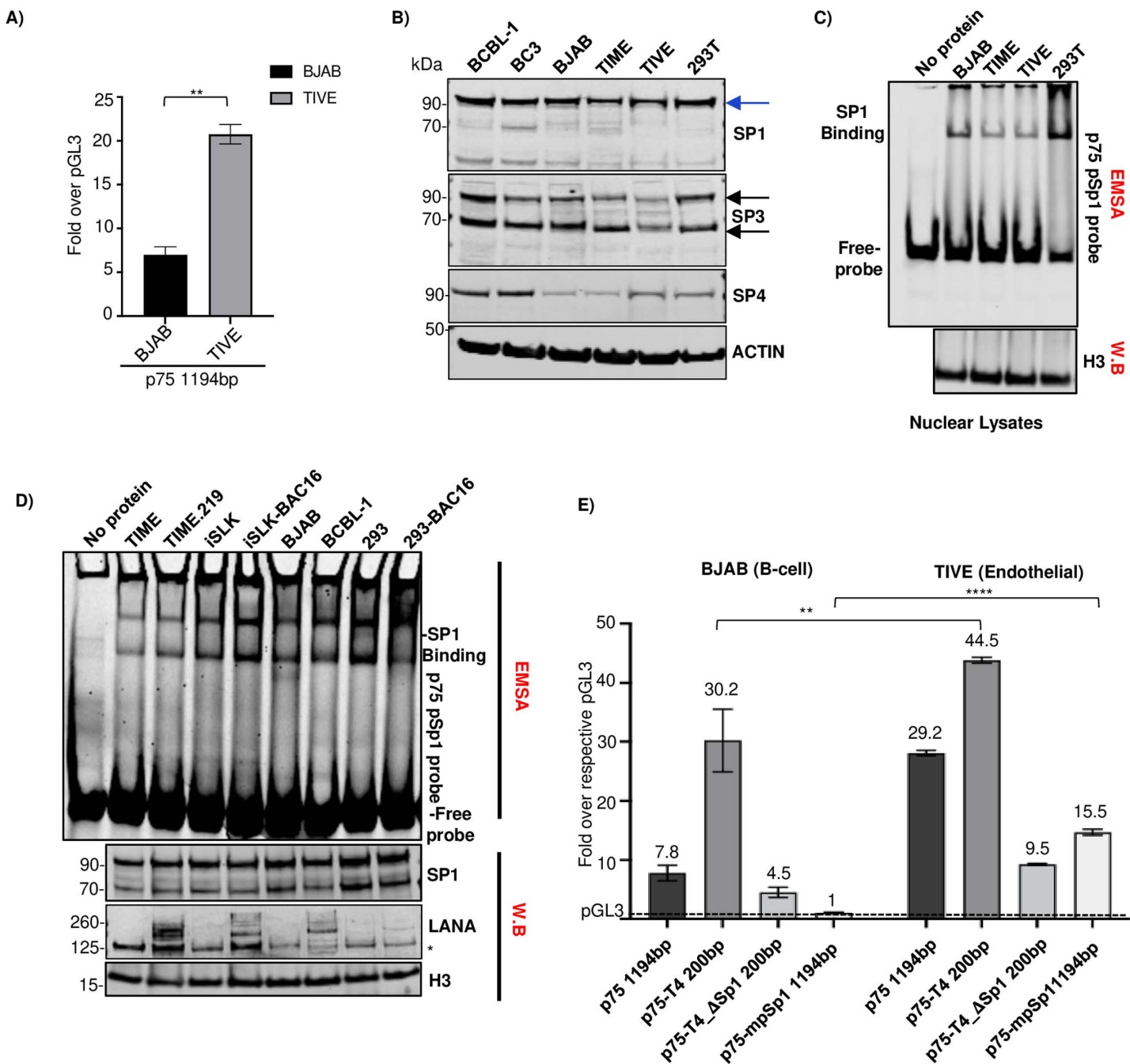

**Fig 7. ORF75 promoter exhibits relatively higher activity in TIVE cells than BJAB cells. A)** Promoter luciferase assay of ORF75 full length promoter in TIVE (endothelial cell line) and BJAB (B-cell line) cells. Results shown are the fold change over the respective pGL3 empty vector for each cell type. Data shown are ± standard deviations of 3 separate experiments. *P*-values (***p* ≤ 0.01) are calculated using two-sided paired *t*-test **B)** Western blot analysis of Sp1, Sp3 and Sp4 protein levels from different cell lines using whole cell lysates. Blue and black arrows indicates full-length Sp1 and alternate SP3 forms, respectively. **C)** EMSA and WB analysis. Nuclear lysates of various uninfected cell lines were analysed for their Sp1 binding activity to the proximal Sp1 element of ORF75 promoter through EMSA. **D)** Same as in C) but with nuclear lysates of KSHV infected and uninfected cell lines. The lower three panels represent parallel WB of the nuclear lysates. *In the LANA WB, the asterisk indicates a non-specific band. **E)** Same as A), except four different ORF75 promoters were used for the promoter luciferase assay in BJAB and TIVE cells. Assayed at 72h post transfection. Numbers on the top of each bar indicates average fold upregulation relative to empty vector pGL3 for each cell type set as 1. Shown are the means ± standard deviations of 3 separate experiments. *P*-values (***p* ≤ 0.01, *****p* ≤ 0.0001, ns not significant) are calculated using two-sided unpaired *t*-test. All blots and gel shift assays were at least replicated in three separate experiments. Blots stripped and reprobed, see S7 Fig for more information.

## Distinct Sp1 protein isoforms and their differential accumulation during KSHV lytic reactivation across cell types

In exploring Sp1 in various cell line, we observed a ~45 kDa Sp1 band in uninduced BCBL-1 and BC3 cell lines (KSHV infected) but not in uninfected BJAB and HUVEC cell lines (Figs 8A and S4E). Further, the 45 kDa Sp1 isoform observed in uninduced PEL cells showed increased accumulation during lytic reactivation of PEL cell line BCBL-1. Also, 68 kDa and 45 kDa forms of Sp1 accumulated in lytic reactivated BCBL-1 cells, the 45 kDa Sp1 form being the prominent form (Fig 8B). In BCBL-1 cells, the appearance of the 45 kDa Sp1 form was associated with a reduction in full length Sp1. As the Sp1 antibody used in our study was a polyclonal antibody towards C-terminus 437-785 aa, we hypothesized that the alternate Sp1 forms might have intact C-terminal DNA binding domains and could still bind to the proximal Sp1 element in the ORF75 promoter. In EMSA analysis, multiple higher order bands were observed in gel shift assay using reactivated BCBL-1 cell lysate with p75 Sp1 DNA probe. Interestingly, a lower order band (red triangle in Fig 8C) was distinctly enriched with lytic reactivation, similar to the alternate form of Sp1 (45 kDa Sp1) as observed in the parallel W.B (Fig 8C). This suggests that the 45 kDa Sp1 isoform might also bind to the proximal Sp1 element of ORF75. Interestingly, in KSHV-infected TIME.219 (endothelial) cells, lytic induction showed accumulation of two alternate Sp1 forms, a ~70 and ~68 kDa. The ~70 kDa form was also observed in uninfected TIME cells upon TPA induction but showed early accumulation in infected TIME.219 cells. The ~68 kDa form was not present in uninduced TIME.219 cells but accumulated in lytic induced TIME.219 cells. (Fig 8D).

## ORF75 protein enhances transcription of itself and other KSHV genes

We next asked whether ORF75 protein could in turn regulate its own promoter activity or that of other KSHV genes. We investigated this by co-expressing a plasmid expressing flag-tagged ORF75 (F-ORF75) protein and a promoter-luciferase fusion construct of ORF75 in 293T cells. Interestingly, we observed a significant increase in ORF75 promoter activity in the presence of F-ORF75 protein (Fig 9A). This effect of F-ORF75 protein was in synergy with RTA protein as co-expressing both F-ORF75 and RTA together further enhanced ORF75 promoter activity (Fig 9B). Because ORF75 is driven by an Sp1 element, we hypothesized that one or more of the Sp1 elements in the ORF75 promoter might play a role in this transcription enhancement mediated by ORF75 protein and that ORF75 may work in tandem with Sp1. Also, it has recently been shown using a CRISPRi tilling assay that the coding region of ORF75 could enhance ORF57 expression [41]. We examined other KSHV genes with a consensus Sp1 element and CCAAT box similar to those of ORF75 (S6A Fig). The promoters of ORF50, K2, and ORF57 all have a consensus Sp1 element sequence and CCAAT box (S6B Fig). We co-expressed a plasmid expressing F-ORF75 protein together with promoter-luciferase constructs of K2 (vIL-6), ORF57, and different lengths of ORF50 (RTA), and ORF73 (LANA). Although the LANA promoter does not have a consensus Sp1 element, it does contain shorter Sp1 binding elements. Three different lengths of the LANA promoter [55] with varying basal expression showed minimal enhancement in the presence of ORF75 protein in our *in vitro* assay (Fig 9C). For both RTA and ORF57, we tested two promoters of varying lengths. Both RTA and ORF57 promoters were highly responsive to ORF75 protein-mediated transcription enhancement (Fig 9D, 9E and 9F). The K2 (vIL6) promoter also showed some level of transcription enhancement (S6C Fig). To verify these observations, we overexpressed F-ORF75 protein in a latently infected 293T-BAC16 cell line and analyzed the RNA expression of several endogenous KSHV genes using RT-qPCR (Fig 9G). An RTA (KSHV-ORF50) expressing construct was used as a positive control and pcDNA3.1 as a negative control. The F-ORF75 expression

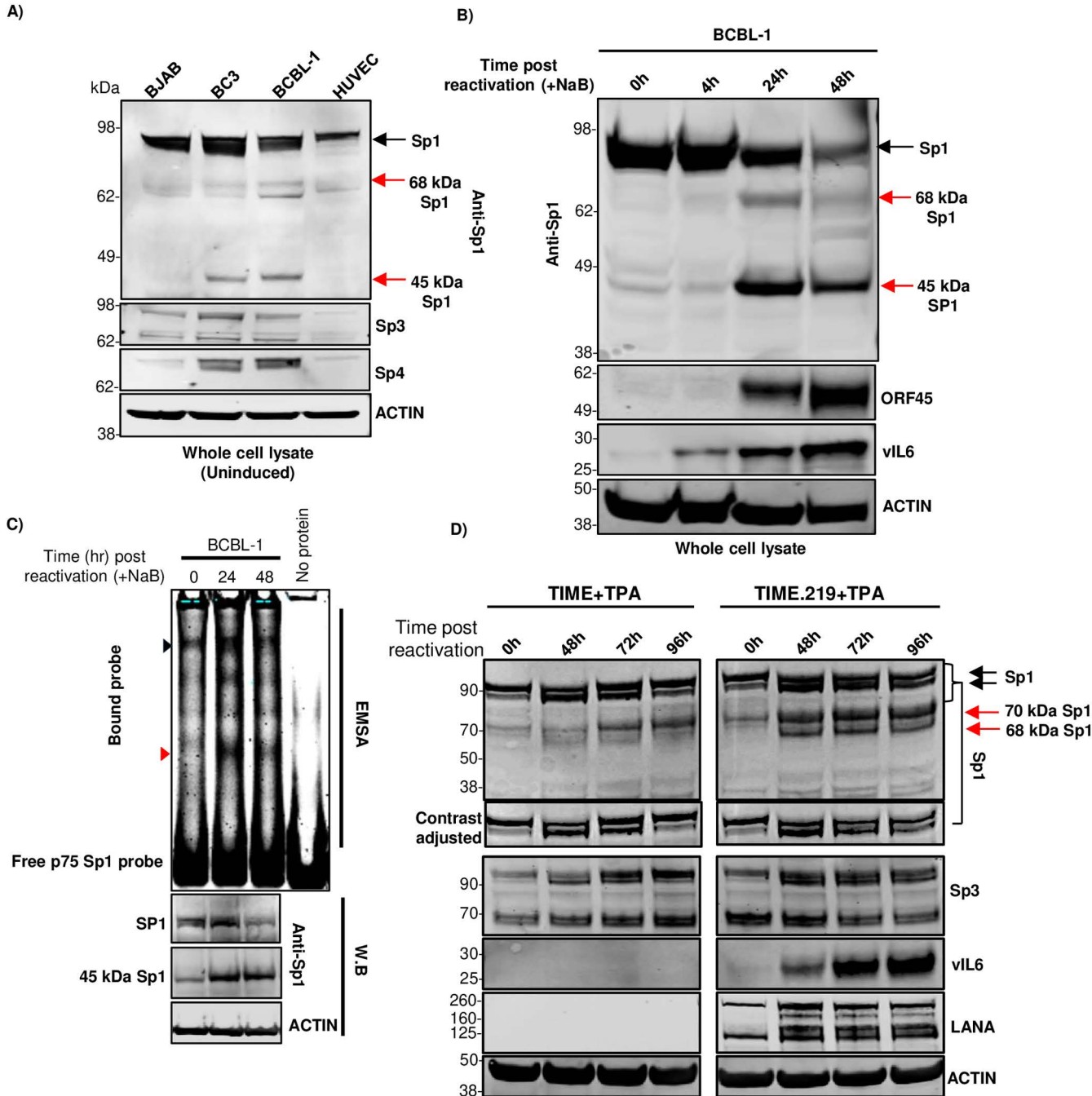

**Fig 8. Degradation of full-length Sp1 protein and accumulation of its variants during KSHV lytic induction.** **A)** WB of whole cell lysate (RIPA buffer) of early passage cell lines showing alternate forms of Sp1 protein in KSHV infected B-cell lines (4-12% BIS-TRIS gel, MES buffer). Sp1, Sp4 and ACTIN are same blot stripped and reprobed. Sp3 blot was run parallel. **B)** WB of BCBL-1 cells at different time points of lytic reactivation blotted with anti-Sp1 antibody. ORF45 and vIL6 were used here as control for lytic reactivation (10% BIS-TRIS gel, MES buffer). Sp1, ORF45 and ACTIN are same blot stripped and reprobed. vIL6 blot was run separately with the same samples. **C)** EMSA and western blot analysis. 10 μg of whole cell lysates (M-PER, Invitrogen) of BCBL-1 cells at different time points of lytic reactivation were analysed for their Sp1 protein binding activity to the proximal Sp1 element of ORF75 promoter through EMSA in a 8% TBE gel. The bottom three panel represent parallel WBs showing accumulation pattern of full-length Sp1 and 45 kDa Sp1 protein isoform during NaB induced lytic induction (10% BIS-TRIS gel, MES buffer). Black and red triangle indicates full-length Sp1 and 45 kDa Sp1 form binding to the proximal Sp1 DNA element of the ORF75 promoter, respectively. Sp1 and ACTIN are same blot stripped and reprobed. **D)** WB of uninfected TIME and infected TIME.219 cell line at different time points of lytic reactivation using TPA. LANA was used as control for infected TIME.219 cells and vIL6 was used as a control for lytic reactivation (8% Bolt BIS-TRIS gel, MES buffer). Black and red arrow indicates full-length and alternate Sp1 forms in WBs, respectively. SP1, vIL6, and ACTIN were on one blot, which was stripped and reprobed, while SP3 and LANA were on a parallel blot, which was also stripped and reprobed. Shown are representative blots, all observations were reproduced in at least 3 separate experiments. See S8 and S9 Figs for full blots.

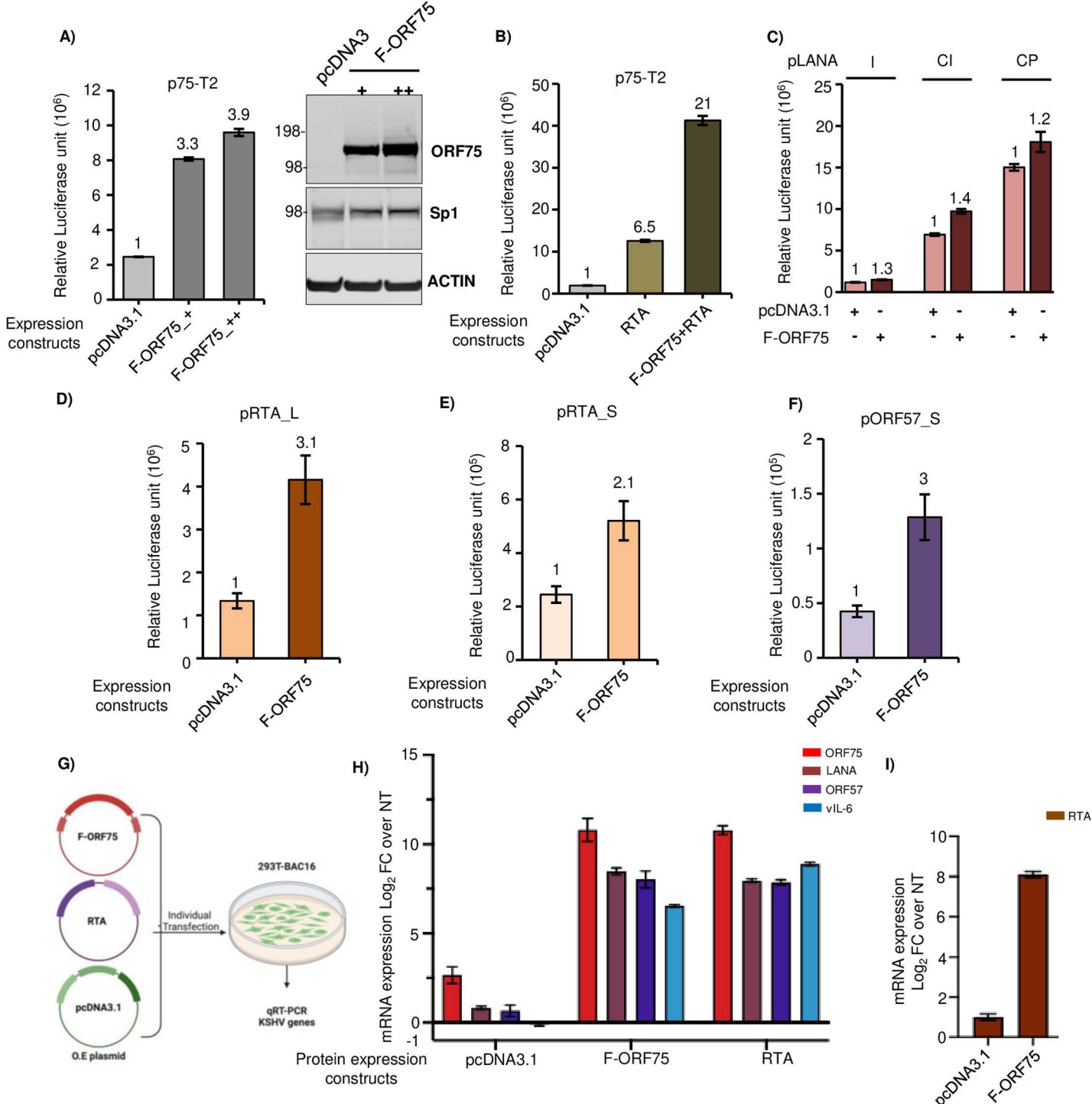

**Fig 9. ORF75 protein enhances expression of immediate early genes. A)** (Left) Promoter luciferase assay of the ORF75-T2 promoter along with co-expression of increasing amount of a plasmid expressing flag-tagged ORF75 (F-ORF75) protein. (Right) Parallel WB analysis of whole cell lysates from the promoter luciferase assay showing F-ORF75 expression. + and ++ indicate 1:2 and 1:4 ratios of ORF75 promoter to protein expression plasmid F-ORF75, respectively. **B)** Same as in A), except RTA was co-expressed alone or together with F-ORF75 along with the ORF75 promoter luciferase construct. **C)** Same as in A), except F-ORF75 was co-expressed along with 3 different length LANA promoter luciferase constructs (LANA I: 570 bp, LANA CI: 1.2 kb, LANA CP: 795 bp. **D and E)** F-ORF75 protein was co-expressed with two different lengths of RTA (ORF50) promoter luciferase fusion constructs. **F)** Same as D), except with an ORF57 promoter luciferase fusion construct. In B-F, 1:4 ratio of ORF75 promoter to protein expression plasmid (F-ORF75 or pcDNA3.1) was used. **G)** Schematic for the assay of KSHV gene expression in 293T-BAC16 cells transfected with F-ORF75, RTA, or a pCDNA3.1 control. Flag-ORF75 and RTA overexpression vectors or a pcDNA3.1 control vector were

transfected into 293T cells latently infected with KSHV followed by qPCR analysis of endogenous KSHV genes (ORF75, LANA, ORF57 and vIL6) 48 hr post transfection. Fig 9G was created in BioRender. **H)** RT-qPCR of select genes in either empty or ORF75 and RTA expression vector transfected cells. Fold change normalized to non-transfected 293T-BAC16 cells. Endogenous ORF75 was detected using a primer pair spanning the 5'UTR and CDS. **I)** Same as H) except that RTA gene expression was analyzed in F-ORF75 transfected 293T-BAC16 cells. Other information: Numbers on the top of each bar in A through F indicates average fold upregulation relative to control set as 1. Error bar indicate ± standard deviations of 3 experiments. pcDNA3.1 plasmid was used as vector control. Promoter luciferase assay in 293T cells was performed at 72h. More information on the different promoters used here is provided in S5D Fig. See S9 Fig for full blots.

construct induced strong induction of endogenous ORF75, ORF50, and ORF57 and lower induction of vIL6 RNA (Fig 9H and 9I). In fact, the F-ORF75 expression plasmid also induced these genes to the same level as RTA, except for vIL6 which was induced 5-fold more by RTA than by F-ORF75. Surprisingly, we observed induction of LANA RNA in F-ORF75-expressing 293T-BAC16 cells even though F-ORF75 showed minimal response towards LANA promoter in promoter luciferase assays. We could not assess endogenous RTA expression in RTA protein over-expressed cells due to the inability to differentiate between the exogenous RTA plasmid and the endogenous RTA sequence in the infected BAC-293T cell. For endogenous ORF75, primer was designed to detect an upstream region present only in the endogenous ORF75 RNA. These findings suggest that the ORF75 coding region can enhance the promoter activity of *ORF50*, *ORF57*, *vIL-6*, and *LANA*, leading to their increased expression.

## Discussion

KSHV ORF75 is classified as a late lytic protein based on studies in PEL cells [36–38]. However, our group and others have recently shown that ORF75 RNA is robustly expressed in KS lesions [42–46]. KS spindle cells are characterized by expression of latent, but not lytic, KSHV genes, so the high expression of ORF75 was unexpected [62–65]. Here, we explore this conundrum and demonstrate that ORF75 is constitutively expressed in endothelial and epithelial cells and that its expression is regulated by Sp transcription factors largely through a proximal Sp element. We further demonstrate that the relatively low constitutive expression of ORF75 in B cells is, at least in part, due to its suppression in these cells, mediated by a distal promoter element. Thus, ORF75 functions as both a latent gene in endothelial and other cells as well as a late lytic gene during KSHV replication (summary Fig 10).

ORF75 is a tegument protein that was initially identified as a structural protein for virion assembly. It was subsequently shown that it also has a role in suppressing innate immunity during initial phase of *de novo* infection by restricting ND-10 function [26,34]. ORF75 RNA and protein levels peak during the late lytic phase in PEL cell lines like BC3 and BCBL-1 and based on these studies it was characterized as a late lytic gene [35]. Gammaherpesvirus lytic promoters are tightly regulated and orchestrated in a unique fashion for temporal expression during the viral life cycle. KSHV late lytic promoters are regulated by vPIC, which currently are known to include six KSHV lytic proteins [27,30]. However, the expression of ORF75 in KS lesions suggested that a different regulatory mechanism induces its expression independent of the lytic cycle.

Since the KSHV-infected cells that define KS lesions are spindle cells with characteristics of endothelial cells, we hypothesized that high latent expression might be cell-specific and occur in endothelial and certain other cells. We found that ORF75 was in fact constitutively expressed in KSHV-infected immortalized endothelial (TIME.219) cells and that the ORF75 promoter was constitutively active in uninfected endothelial and epithelial cells. Exploring this further, we found substantial evidence that ORF75 was regulated by Sp proteins binding to proximal Sp1-like elements and distal CCAAT boxes. Pascal et al., (1991) showed that binding of Sp1 to a proximal and distal Sp1 element simultaneously constitutes a synergistic

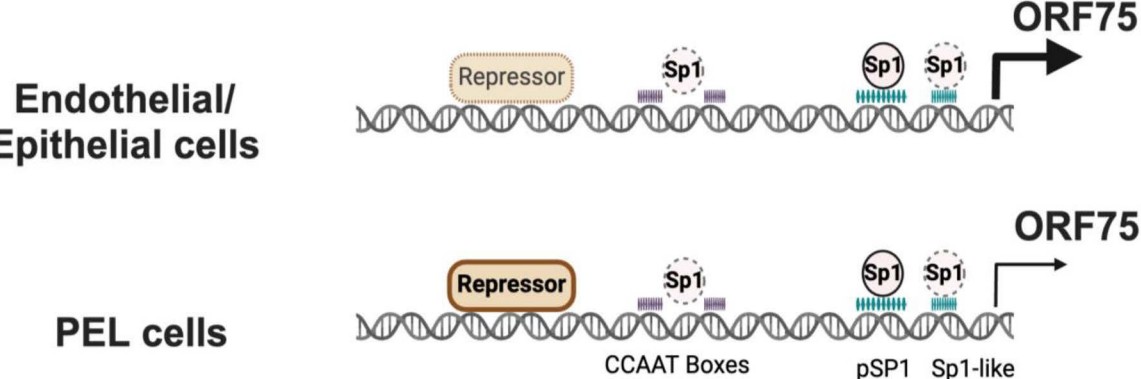

Created in https://BioRender.com

**Fig 10. Summary schema.** Simplified summary schema showing the regulation of ORF75 promoter's basal transcription activity. The activity is primarily regulated by the binding of the Sp1 protein to Sp1 elements. There is evidence that in all cell lines, the Sp1 complex interacts with the proximal Sp1 element, the Sp1-like elements, and the CCAAT boxes to activate this transcription. Also, a repressor, whose identity is unknown but acts via the distal region of the ORF75 promoter, variably inhibits basal transcription. This repressor's effect determines the promoter's activity across different cell lines, even in the presence of Sp1 activation. In B-cells, the distal region of the ORF75 promoter strongly represses its activity, leading to lower ORF75 transcription during latency. By contrast, endothelial and epithelial cells experience less repression and more activation of the ORF75 promoter due to the proximal Sp1 element. This results in higher ORF75 transcript levels, even though there is no lytic viral DNA replication. This figure was created in BioRender.

Sp1 activation complex [66], and our data suggests that formation of this complex might affect ORF75 expression in the different cell types.

While we found that ORF75 promoter had some constitutive activity in B cell lines, its activity was substantially lower than in endothelial and epithelial cell lines. We initially hypothesized that this might be because the B cell lines had low levels of Sp1. Somewhat surprisingly, however, the B-cell lines were found to have high expression of Sp proteins and showed equivalent ORF75 promoter Sp binding (Fig 7). This suggested that the ORF75 promoter might be under active repression by another transcription factor. This hypothesis was strengthened by our observation that a truncated ORF75 promoter (p75-T4) showed higher transcriptional activity in both a B-cell and an endothelial cell line, suggesting that ORF75 promoter activity was repressed by a distal promoter element absent in the truncated promoter in both the cell types. The promoter activity of the truncated promoter vs full length promoter was substantially higher in B-cells than in endothelial cells, providing evidence for substantially stronger repression in B-cells. We further showed that BCL-6 protein, a B-cell specific transcription factor, can actively repress ORF75 promoter in 293T cells (S5A Fig). However, BCL6 suppressed both the full-length and the truncated promoter, and PEL cell lines show little to no expression of BCL6 during latency [67], so its role in suppression of ORF75 in PEL lines remains unclear.

We also observed accumulation of alternate isoforms of Sp1 protein during induction of lytic cycle in both PEL cells and TIME.219 cell line. In the PEL cells, a 45 kDa Sp1 isoform was observed in latent uninduced cells, which increased during lytic induction. 45 kDa and 68 kDa Sp1 forms were reported previously in a Burkitt lymphoma cell line (BL-60) [68]. Formation of Sp1 alternate forms has been attributed to caspase cleavage as well as splicing [68–70]. As seen here, these alternate forms have an intact DNA binding domain and can bind Sp1 element of the ORF75 promoter (Fig 8A, 8B and 8C). Sp1 alternate form have been reported to act as dominant negative mutants to modulate its activity [71]. Interestingly, the Sp1 protein

isoforms observed in lytic reactivation of the TIME.219-infected endothelial cell line was different from that of PEL cells with 68 kDa Sp1 isoform being prominent. The difference in the accumulation of Sp1 isoforms in various cell lines along with different co-transcription factors could lead to differential regulation of Sp1 protein targets and thus may contribute to variations in ORF75 expression. Additional studies are needed to sort out these complexities.

It has recently been shown that ORF75 is essential for KSHV lytic DNA replication and is also essential for or facilitates transcription of various immediate early or early genes [26,30,41]. These vital functions would require its presence during early stages of KSHV lytic replication, and the finding here of constitutive expression during latent infection of endothelial cells is consistent with this function. And as we have shown here, while ORF75 has been classified as a latent gene in PEL, there is also some constitutive production in B cells that could suffice to enable early KSHV gene transcription. The question remains as to whether its function in enabling early gene transcription might be through a non-coding activity of ORF75 RNA. However, this seems unlikely, as ORF75-STOP virus, lacking ORF75 protein but expressing ORF75 RNA, failed to support viral genome replication [26]. However, additional studies will be needed to clarify these issues.

Similar to KSHV ORF75, BNRF1, the EBV homolog of KSHV ORF75, is essential for B-cell lymphomagenesis and the induction of latency. Without BNRF1, SMC proteins silence the EBV genome before induction of latency. BNRF1 RNA was also shown to be expressed during EBV latency, even though it is generally characterized as a lytic gene. Others have proposed various functions of BNRF1 during latency, which also involve the induction of various latent genes [48,49]. And similar to the ORF75-mediated induction of KSHV RTA, BNRF1 has been shown to promote the expression of EBV transactivator protein ZTA [72].

As seen here, in addition to being essential for lytic KSHV activation and enhancing expression of a number of lytic KSHV genes [26,30,41], ORF75 protein has a feedback regulation of its own promoter, enhancing its own promoter's transcriptional activity by multiple fold. Through its ability to activate various KSHV gene promoters, it thus plays a key role in regulation of KSHV lytic replication. Given this, it is interesting to speculate as to why the virus has evolved to have lower constitutive expression in B cells than in endothelial or epithelial cells. B-cells are generally considered the primary reservoir for latent KSHV, and the relatively low latent expression of ORF75 in B cells may serve to tilt the balance in these cells towards latency. On the other hand, KSHV virus is predominantly transmitted through saliva, in part from infected epithelial cells [73,74], and the high constitutive activity of ORF75 promoter in epithelial cells may help prime these cells for lytic replication.

In summary, we show here that the high expression of ORF75 in KS lesions is largely due to its substantial constitutive expression in endothelial cells and that this is mediated by Sp1 protein through a proximal Sp1 promoter element along with two distal CCAAT boxes. The lower constitutive expression of ORF75 promoter in B cells is primarily because of a suppressive factor for ORF75 in these cells. Given the increasing appreciation of the key roles of ORF75 in the KSHV life cycle, it will be important to further clarify the factors regulating its expression and its function in regulating itself and other KSHV genes. Finally, given its central roles in the KSHV life cycle, it may be worth considering therapies targeting ORF75 for the treatment of KSHV-associated diseases.

## Methods

### Ethics statement

Biopsy samples of KS lesions were obtained from patients on a tissue procurement protocol in the HIV and AIDS Malignancy Branch (NCT00006518). This protocol (local protocol number

01-C-0038) was approved by the National Institutes of Health Institutional Review Board. All patients provided written informed consent prior to any study procedures in accordance with the Declaration of Helsinki.

## Cell lines and culture conditions

The PEL cell lines BCBL-1 (National Institutes of Health, AIDS Research and Reagent Program, Rockville, MD) and BC-3 (ATCC, Rockville, MD), which harbor KSHV only, and the KSHV-uninfected B cell line cell line BJAB (ATCC, Manassas, VA) were grown in RPMI 1640 media (Invitrogen, Waltham, MA) supplemented with 15% heat-inactivated (HI) fetal bovine serum (FBS) (Thermo Scientific, Waltham, MA) and 1% Pen-strep glutamine (PS) (Invitrogen) at 37°C, 5% $CO_2$ under atmospheric oxygen conditions. HEK-293T (ATCC, Manassas, VA), a human embryonic kidney cell line, and HepG2, a human hepatoma cell line (ATCC), were maintained in Dulbecco's modified Eagle's medium (DMEM) (Invitrogen) supplemented with 10% FBS and 1% PS at 37°C, 5% $CO_2$. HEK-293T cells infected with KSHV (BAC16) were a kind gift from Dr. Ting-Ting Wu, University of California Los Angeles. These cells were maintained in the same media and conditions as uninfected cells but with an addition of 100 μg/ml hygromycin B (Invitrogen). iSLK cells and iSLK-BAC16 were a kind gift from Dr. Jae Jung, Case Western Reserve University. Cells containing rKSHV-BAC16 virus were cultured in DMEM supplemented with 10% tetracycline-free FBS, 1% PS, 1μg/ml puromycin, and 250 μg/ml G418 at 37°C, 5% $CO_2$ and normoxic $O_2$. Additionally, 1 mg/ml hygromycin was added only in iSLK-BAC16. The SL2 insect cell line (ATCC) was cultured in Schneider's Drosophila medium (Thermo Scientific) supplemented with 10% HI-FBS and 1% PS at 26°C atmospheric $CO_2$ and $O_2$. HDMVEC cells (ATCC) were cultured in vascular cell basal medium (VCBM, ATCC-PCS-100-030) supplemented with the microvascular endothelial cell growth kit-VEGF (ATCC-PCS-110-041) at 37°C, 5% $CO_2$ and normoxic $O_2$.

## RNAscope and IHC

Human herpesvirus 8 (HHV8) ORF75 RNA expression was detected in these KS biopsies by staining 5 μm FFPE tissue sections with RNAscope 2.5 LS Probes V-HHV8-ORF75 (ACD, Cat# 562058) using the RNAscope LS Multiplex Fluorescent Assay (ACD, Cat# 322800) with a Bond RX auto-stainer (Leica Biosystems) Multiplex Fluorescent Assay (ACD, Cat# 322800) using the Bond RX auto-stainer (Leica Biosystems) with a tissue pretreatment of 30 minutes at 100°C with Bond Epitope Retrieval Solution 2 (Leica Biosystems) and 1:750 dilution of OPAL 570 (AKOYA, Biosciences). Immunohistochemistry for HHV8 LANA protein was performed after RNAScope using a mouse anti-human HHV8_LNA, clone 13B10 antibody (Leica Biosystems, cat# NCL-L-HHV8-LNA) at a 1:50 dilution for 30 minutes using the Bond Polymer Refine Kit (Leica Biosystems) minus DAB and Hematoxylin with a 1:750 dilution of OPAL 690 reagent for 30 min. The RNAscope 2.5 LS Negative Control Probe (*Bacillus subtilis* dihydrodipicolinate reductase (*dapB*) gene, cat# 312038) followed by IHC with no primary antibody was used as an ISH and IHC negative control. The RNAscope LS 2.5 Positive Control Probe Hs-PPIB (cat# 313901) was used as a technical control to ensure the RNA quality of tissue sections was suitable for staining. Slides were digitally imaged using an Aperio ScanScope FL Scanner (Leica Biosystems).

## Generation of TIME.219 and Sp1-Kd 293T cell lines

TIME cells (ATCC) were cultured in vascular cell basal medium (VCBM, ATCC-PCS-100-030) supplemented with the microvascular endothelial cell growth kit-VEGF (ATCC-PCS-110-041). TIVE cells [61] (gift from Dr. Rolf Renne, University of Florida)

were maintained in DMEM supplemented with 10% FBS and 1% PS. For the generation of KSHV latently-infected cell lines, early passages of TIME were seeded at $1 \times 10^5$ cells/well of a 12-well plate in complete growth medium with antibiotics. The next day, the growth medium was removed and replaced with complete medium without antibiotics. The cells were infected with rKSHV.219 virus at MOI 1 in the presence of 5 μg/ml polybrene. The plate was centrifuged at 500 g for 1 hr. at room temperature (RT) to enhance viral attachment. After centrifugation, the plate was incubated for 2 hr, following which the virus-polybrene medium was replaced with fresh medium. Six days post-infection, the cells were selected with 5 μg/ml puromycin for infected TIME cells. The cells were subcultured twice per week, with the final puromycin concentration being 7 μg/ml for infected TIME cells. The resulting TIME.219 cells were subcloned at 5 to 10 cells/well of a 96-well plate. Two clones for each cell line were selected based on uniform GFP expression among the cells within a cloned population.

To generate shRNA-mediated Sp1 knockdown (Sp1-Kd) 293T cell lines, 293T cells were plated in a 6-well plate at a density of $3\times10^5$ cells per well in DMEM supplemented with 10% FBS and 1% penicillin-streptomycin-glutamine. Twenty-four hours after seeding, the media were replaced with 2 mL of fresh DMEM without antibiotics.

Next, 20 μL of Sp1 shRNA lentiviral particles (sc-29487-V, Santa Cruz) and 20 μL of control shRNA lentiviral particles (sc-108080, Santa Cruz) were each mixed with polybrene at a final concentration of 4 μg/mL. The mixtures were incubated with the cells for 16 hours.

72 hr post-infection, selection was performed using 1 μg/mL puromycin. Two single colonies were isolated and expanded to establish the Sp1-Kd and control 293T cell lines.

## Cloning and plasmids

The full-length ORF75 promoter pGL3-p75 utilized in this study contains 1194 bp spanning positions 134461 to 135654 with reference to the KSHV genome NC_009333. The promoter region was amplified by specific PCR primers containing KpnI and XhoI sites and cloned into the pGL3 basic vector (Promega). Various mutations and truncations of the pGL3-p75 promoter regions were synthesized as DNA fragments using Geneart DNA synthesis services or Genscript services, followed by cloning into pGL3 basic vector using KpnI and XhoI sites. Plasmids were purified using Qiagen plasmid preparation kits. For the full-length ORF75 protein coding construct, the ORF75 sequence was amplified using specific primers containing NotI and SalI. Another fragment containing a linker (GGGSGGGS) and the 3X-Flag sequence was synthesized with SalI and XbaI on either end. A three-fragment ligation was performed with pcDNA3.1 vector digested with NotI and XbaI, ORF75 fragment, and linker-flag tag fragment to obtain the pcDNA3.1::ORF75-FLAG construct.

Expression plasmids for HIF-1α and HIF-2α mutant in pcDNA3.1 vector backbone expressing degradation-resistant forms contain P405A, P531A mutations, have been described previously [31]. RTA-expressing plasmid (pcDNA-RTA) was a kind gift from Keiji Ueda, Osaka University Graduate School of Medicine, Osaka, Japan. LANA-expressing vector was cloned into pCMV-Tag2B vector as previously described [75]. Mammalian Sp protein expression plasmids (pN3-Sp1FL (24543), pN3-Sp3FL (24541), pN3-Sp4FL (107719) and empty pN3 plasmid (24544)) were a kind gift from Dr. Guntram Suske and obtained through Addgene services. The insect cell line expression plasmids were a kind gifts from Dr. Robert Tijan through Addgene (pPAC-Sp1 (12095), pPAC0 (12094)). Control Beta-galactosidase for insect cell line transfection was obtained through Addgene (pAC5.1-bGAL-V5HIS, 136238). Mammalian lentiviral expression plasmids used for transient expression in promoter assays were a kind gift from Dr. Feng Zhang through Addgene (control pLX_TRC317-eGFP (TFORF3549, 145025), pCXN2-BCL6 (40346), pLX-OCT2 (143550)).

## Transfection and reporter assay

Reporter luciferase assays were performed in several cell types. For 293T, SLK, Cos7, HepG2 and 293T-BAC16 cell lines, $3x10^5$ cells were seeded in a 6-well plate the day before transfection. On the following day, media was replaced, and cells were transfected with 400 ng of promoter luciferase construct or cotransfected with a 1:4 ratio of promoter luciferase to expression plasmid using Fugene 6 (5:1 transfection reagent to DNA, Promega). pcDNA3.1 empty vector was used as carrier DNA during transfection. 70 ng of β-Gal expression plasmid was used as a transfection normalization control. TIVE cells were either transfected using transfeX (ATCC) transfection reagent or with nucleofection. For transfeX based transfection, $3x10^5$ cells were plated in a 6-well plate the day before transfection. 600 ng of promoter construct and 400 ng of β-Gal plasmid or nano-luciferase transfection control plasmid was diluted in 200 μl of Opti-mem media and transfeX reagent was added in 3:1 ratio of transfection reagent to DNA (3 μl in 200 μl mixture). For nucleofection, 1 μg of promoter luciferase reporter construct along with 600 ng of β-Gal plasmid or nano-luciferase plasmid was nucleofected with nucleofector IIb using program A-034 and KIT V into $4 \times 10^6$ cells. SL2 cells were also either nucleofected or transfected using TransfeX reagent. $5 \times 10^6$ cells were nucleofected with 1 μg of promoter construct, 0.5 μg of β-Gal control plasmid, 1 μg of pPAC-Sp1 plasmid and pcDNA3.1 carrier plasmid to a final plasmid concentration of 4 μg. Nucleofector IIb using program G-030 and KIT V. For TransfeX-based transfection, $2x10^6$ cells were seeded in a 6-well plate. The following day, media was carefully replaced without disturbing the loose layer of adherent cells. 350 ng promoter construct, 350 ng pPAC-Sp1 or pPAC0 and 300 ng of β-Gal plasmid was diluted in 200 μl of Opti-mem along with 3 μl of transfeX reagent. BJAB cells were transfected with nucleofector 2II using program C-009 and KIT V or program T-016 with B-cell nucleofection kit. For BJAB cell nucleofection, 3-4 $x10^6$ cells were transfected with either 4 μg of promoter luciferase and 3 μg of β-Gal with B-cell nucleofection kit or 2.7 μg of promoter and 300 ng of NanoLuc control plasmid with kit V. Irrespective of the cell type, transfection was always performed in healthy early passage cells with >97% live cells at the beginning of transfection.

Transfected cells were incubated for 72 hr. unless otherwise mentioned. Cells were carefully washed with PBS and lysed in reporter lysis buffer (RLB, Promega) containing protease inhibitors (PI) (HALT protease, Thermo). A quick freeze-thaw was also used to attain complete lysis followed by centrifugation and supernatant separation. Luciferase readout was performed using 20 μl of the supernatant lysate using a multimode plate reader with a 10-second readout. Trans-effect of the secondary control plasmid, if observed, was minimized by increasing the difference in the amount of test and transfection control plasmids as recommended by the manufacturer (Promega). All transfections except SL2 transfection were verified using at least 2 different normalization methods that included β-galactosidase assay, Renilla luciferase, NanoLuc assay.

## Protein extraction and western blotting

Whole cell lysates were prepared using RIPA lysis buffer (Thermo Scientific) supplemented with 2X PI (Roche, Complete tablets), 1X PhosStop (Sigma-Aldrich), and 20 μM NEM. Briefly, 10 cell volumes of chilled lysis buffer were added to the cells followed by thorough vortexing and ice incubation cycles (20 min). The samples were also sonicated for 30 sec to reduce viscosity followed by centrifugation at 13K RPM for 30 min at 4°C. Nucleo-cytoplasmic extraction was done according to the manufacturer's protocol (NE-PER, Thermo Scientific). Gels for western blotting were run in the Xcell Surelock electrophoresis system using MES buffer (unless otherwise mentioned) and BIS-TRIS NuPAGE gel (Invitrogen) at a constant

200V. Transfer to a nitrocellulose membrane was performed using a semi-dry apparatus iBlot3 (Invitrogen). The blots were blocked with intercept TBS blocking buffer (Licor) for 1 hr at RT, primary antibody incubation was either 1 hr at RT or overnight at 4°C followed by a fluorescent secondary anti-goat rabbit or mouse (IRdye 700, 800, Licor) incubation at RT for 1 hr. Washes were performed in TBST buffer, and the blots were scanned using an Odyssey scanner. Blots were stripped using nitro 5X (Licor) stripping buffer for further reprobing. Antibodies used in the study are listed in S1 Table.

### Electrophoretic mobility shift assay (EMSA)

EMSA was performed using IRdye-700 labeled dsDNA probe (5' labeling) (IDT). In a 20 μl reaction, 2.5 nM labeled probe was incubated at RT for 25 min with nuclear lysates or whole cell lysates in the presence of binding buffer (Tris-Cl pH 8 25 mM, 100 mM NaCl, 1 mM EDTA, Glycerol 5%, Poly (dI: dC), 0.1% Tween 20, DTT 3 mM). Nuclear lysates were prepared using the NE-PER nucleo-cytoplasmic extraction kit, and whole cell lysates were made using M-PER lysis buffer (Thermo Scientific) supplemented with PI. A glycerol-based Orange G dye was used for loading the reaction mixture onto 8% TBE gels. Gels were resolved at a constant 65V using pre-chilled 1% TBE buffer under cold conditions. Resolved gels were directly scanned using an Odyssey scanner. For supershift assays, antibodies were added to the reaction mixture 15 min after addition of DNA probe and protein. Total reaction time was 45 min for supershift assays. For competition assays, the cold competitive probes were pre-incubated with the labeled probe and then added to the reaction mixture.

### ChIP assay

Chromatin immunoprecipitation assay was performed as previously described in [76] using Abcam ChIP kit (ab500). Briefly, $3\times10^6$ cells from latent iSLK-BAC16 cells (cultured with all the required antibiotics) were collected, washed with PBS twice, and fixed with 1.1% formaldehyde for 10 min at RT. Fixed cells were lysed in buffer C followed by reaction stop with buffer D and 12 min sonication (cycle 25 sec on/off, 4°C) using a Diagenode Bioruptor. 10 μg of Sp1 antibody (Proteintech, 21962-1-Ap), Sp3 antibody (Proteintech, 26584-1-Ap), IgG antibody (CST), and H3 antibody (Abcam, ChIP grade) were incubated with the sheared chromatin overnight at 4°C in a rotary shaker. Bound antibodies were purified using Protein A agarose beads followed by thorough washing to remove any non-specific binding. Bound DNA was purified using DNA purifying slurry included in the kit as per manufacturer's protocol. The ChIP samples were used for quantitative PCR using specific primers (S2 Table) and qPCR was performed in QuantStudio3 (Invitrogen).

### RNA isolation and qRT-PCR

Total cellular RNA was isolated with TRIzol reagent (Invitrogen) using Direct-zol RNA miniprep kit (Zymo research). $3\times10^6$ cells were pelleted, washed with PBS, and resuspended in 800 μl of TRIzol reagent. The mixture was vigorously vortexed and loaded to an RNA spin column. Bound RNA was washed followed by an on-column DNaseI digestion. RNA was eluted in ultrapure water, and concentration was measured using nano-drop (Thermo Scientific). 2 μg of the purified RNA was submitted to a second DNaseI digestion in solution using Turbo DNase (Thermo Scientific). Purified RNA was used for cDNA synthesis using SuperScript III (Thermo Scientific). Power SYBR green PCR master mix (Thermo Scientific) was used for q-PCR reaction. The cycle conditions were as follows: initial denaturation and enzyme activation 95°C for 10 min, cycle denaturation 95°C for 15 sec, annealing for 1 min at 60°C, followed by melt curve analysis. Primers for qPCR are listed in the S2 Table.

## Statistical analyses

Statistical analyses were performed using GraphPad Prism v.10.2.3. The significance of mean differences was determined using two-sided paired Student's *t*-test for promoter analysis in same cell type. Two-sided unpaired *t*-test was used to compare same promoter activities in different cell types. Error bars shown in graphical data represent mean ± SD. A two-tailed value of $P < 0.05$ was considered statistically significant.

## Supporting information

**S1 Fig.  Dual stained zoomed-in KS skin tissue region.** A) KS Skin lesion zoomed in section from panel A, showing colocalization of ORF75 RNA and LANA protein.
(TIF)

**S2 Fig.  KSHV gene expression analysis in a infected epithelial and a B-cell line.** A) qPCR analysis of representative genes of latent and lytic cycle in latently infected iSLK-BAC16 and PEL cell line, BC3. N=3 biological replicate with three qPCR technical replicate. Expression of all genes were normalized to respective LANA expression for each cell type. Internal reference gene GAPDH. **B)** qPCR analysis of ORF75 RNA, K15 RNA using two distinct primer set targeting exon 1 and 3 of the K15 gene and ORF57 RNA in latently infected TIME.219, BCBL-1 and ISLK-BAC16 cells. **C)** qPCR analysis of *de novo* infection of primary Human Dermal Microvascular Endothelial cells (HDMVEC) with rKSHV.219 virus at MOI 1. Samples were collected at 3 and 13 days post infection. ORF72 and PAN RNA serves as a marker for latent and lytic infection cycle, respectively. Shown are the means ± standard deviations of at least 3 separate experiments.
(TIF)

**S3 Fig.  Basal ORF75 promoter activity is independent of HRE and ARE elements.** A) Table showing positions and lengths of different KSHV gene promoters used in Figs 3D and S3B. **B)** Promoter luciferase assay of ORF75 and ORF74 (1.2 kb) promoter in HEK293T and HepG2. Data normalised to respective pGL3 vector in each cell line. Histogram represents mean with SD as error bars for three biological replicate. Assayed at 72h post transfection. **C)** Promoter luciferase assay of ORF75 promoter in HepG2, SLK and COS-7 cells. Assayed at 72h post transfection. ORF75 promoter activity was normalized to pGL3 vector control activity set as 1 for each cell line. **D)** Promoter luciferase assay of the ORF75 promoter in HepG2 cells, coupled with transient overexpression of NRF2, HIF1α, and HIF-2α degradation-resistant mutants. X and 2X indicate 1:2 and 1:4 ratios of ORF75 promoter to protein expression plasmid, respectively. **E, F, G)** Promoter luciferase assay of ORF75 promoter in HepG2 cells with various treatments. All treatments were done 24h post transfection. Assayed at 48h. Histogram for all except B) is one experiment.
(TIF)

**S4 Fig.  ORF75 promoter's proximal Sp1 element is regulated by Sp1 transcription factor.** A) EMSA showing binding of Sp. proteins with dsDNA probe of ORF75 promoter along with a positive control Sp1 probe (Li-COR, P/N: 829-07926) in a 8% native PAGE gel. Three different protein concentrations used 10, 20 and 30 μg. **B)** Same as in A), except only p75-Sp1 probe used with three different concentrations of 293T whole cell lysates (10, 25 and 50 μg). **C)** ChIP assay analysing Sp1 and Sp3 binding regions of ORF75 promoters in latent iSLK-BAC16 cells. Binding was analysed by chromatin immunoprecipitation followed by qPCR. The schematic on the top shows the location of the various primers used to detect Sp protein abundance on the ORF75 promoter. H3 was used as a positive ChIP control and IGG antibody as isotype control. R1 and R2 are two separate experiments. **D)** Same as in C) except a 68 bp long DHFR

gene promoter region located at -400 bp from transcription start site of the human DHFR gene was used as a positive control for Sp protein binding **E)** Western blot analysis of Sp1, Sp3 and Sp4 protein levels from different cell lines using whole cell lysates. Black and red arrow indicates full-length Sp1 and alternate SP1 forms in W.B, respectively. See S7 Fig for full blots.
(TIF)

**S5 Fig. Promoter repressor assay.** A) Promoter luciferase assay of full length ORF75 (p75) and truncated promoter construct (p75-T4) with co-expression of BCL6 expression constructs in 293T cell line. pcDNA3.1 plasmid was used as vector control for BCL6 expression plasmid. Assayed at 3 days post transfection. Error bar indicate ± standard deviations of 3 experiments.
(TIF)

**S6 Fig. Multiple consensus Sp1 elements are present across KSHV genome. A)** Table showing multiple consensus Sp1 element position throughout KSHV genome. Only complete consensus sequence is shown here. Consensus sequence KGGGCGGRRY, where K stands for G or T and R stands for G or A. **B)** Schematic diagram of various KSHV gene promoters used in this study. **C)** Promoter luciferase assay of vIL6 promoters along with co-expression of F-ORF75 protein in HEK293T cells. Assayed at 72h post transfection. Shown are the means ± standard deviations of 3 separate experiments.
(TIF)

**S7 Fig. Complete, uncropped EMSA and western blots.** Figure labels used for all uncropped blots here are same as the figure labels used in the cropped blots. 680 and 800 nm indicates the *LICOR* IR dye channel used for scanning. Indicated blot number can be used to trace stripping and reprobing order.
(TIF)

**S8 Fig. Complete, uncropped EMSA and western blots, part 2.** Figure labels used for all uncropped blots here are same as the figure labels used in the cropped blots. 680 and 800 nm indicates the *LICOR* IR dye channel used for scanning. Indicated blot number can be used to trace stripping and reprobing order.
(TIF)

**S9 Fig. Complete, uncropped EMSA and western blots, part 3.** Figure labels used for all uncropped blots here are same as the figure labels used in the cropped blots. 680 and 800 nm indicates the *LICOR* IR dye channel used for scanning. Indicated blot number can be used to trace stripping and reprobing order.
(TIF)

**S1 Table. List of antibodies used in the study.**
(DOCX)

**S2 Table. List of primers used in the study.**
(DOCX)

**S3 Table. Experimental data for Figures.**
(XLSX)

## Acknowledgments

The authors thank the patient-volunteers who contributed biopsy specimens and the clinical staff of the HIV and AIDS Malignancy Branch and the NIH Clinical Center. We also thank Dr. Joseph M. Ziegelbauer of the HIV and AIDS Malignancy Branch and other members of the Yarchoan laboratory for helpful discussions.

## Author contributions

**Conceptualization:** Ashwin Nair, Robert Yarchoan.

**Formal analysis:** Ashwin Nair.

**Investigation:** Ashwin Nair, Andrew Warner, Baktiar Karim, Ramya Ramaswami.

**Methodology:** Ashwin Nair, Andrew Warner, Baktiar Karim.

**Resources:** Robert Yarchoan.

**Supervision:** David A Davis, Robert Yarchoan.

**Writing – original draft:** Ashwin Nair, Robert Yarchoan.

**Writing – review & editing:** David A Davis, Andrew Warner, Baktiar Karim, Ramya Ramaswami, Robert Yarchoan.

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
