## [Decision Letter · Decision Letter 0]

29 Oct 2024

PPATHOGENS-D-24-02092The Elevated Expression of ORF75, a Lytic KSHV Gene, in Kaposi Sarcoma Lesions is Driven by a GC-rich DNA cis Element in its Promoter RegionPLOS Pathogens Dear Dr. Yarchoan, Thank you for submitting your manuscript to PLOS Pathogens. After careful consideration, we feel that it has merit but does not fully meet PLOS Pathogens's publication criteria as it currently stands. Therefore, we invite you to submit a revised version of the manuscript that addresses the points raised during the review process. Please submit your revised manuscript within 60 days Dec 28 2024 11:59PM. If you will need more time than this to complete your revisions, please reply to this message or contact the journal office at plospathogens@plos.org. Please include the following items when submitting your revised manuscript:* A rebuttal letter that responds to each point raised by the editor and reviewer(s). You should upload this letter as a separate file labeled 'Response to Reviewers '. This file does not need to include responses to any formatting updates and technical items listed in the 'Journal Requirements' section below.* A marked-up copy of your manuscript that highlights changes made to the original version. You should upload this as a separate file labeled 'Revised Manuscript with Track Changes '.* An unmarked version of your revised paper without tracked changes. You should upload this as a separate file labeled 'Manuscript '. If you would like to make changes to your financial disclosure, competing interests statement, or data availability statement, please make these updates within the submission form at the time of resubmission. Guidelines for resubmitting your figure files are available below the reviewer comments at the end of this letter. We look forward to receiving your revised manuscript. Kind regards, Dirk P. Dittmer, Ph.D.Academic EditorPLOS Pathogens Robert KalejtaSection EditorPLOS Pathogens Michael Malim

Editor-in-Chief

PLOS Pathogens

orcid.org/0000-0002-7699-2064   **Journal Requirements:** **Additional Editor Comments (if provided):** Note that reviewer two added his comments as a separate file attached herewith.It will be important to address the concerns of rigor, robustness and reproducibility raised by reviewer one explicitly.**Reviewers' Comments:** Reviewer's Responses to Questions

**Part I - Summary**

Reviewer #1: Nair et al. show that KHSV ORF75, an annotated lytic protein, is pervasively expressed in latently infected KS skin lesions. They examine the distinct behavior of ORF75’s promoter in several different cell lines, demonstrating that its activity is most prominent in endothelial and less so in B cells. This may be due to differences in expression or activity of the SP1 transcription factor between cell types, and their promotor deletion mapping experiments identify key SP1 binding sites within the ORF75 promoter and suggest the presence of a repressive element that is active in B cells. The combined promoter deletion mapping experiments in mammalian cells and SP1 complementation assays in SP1-deficient drosophila cells nicely support an SP1-dependent mechanism. They also pursue SP1 gel shifts and overexpression experiments in mammalian cells (these are somewhat hard to interpret given that it’s not clear that SP1 is limiting—it could be more informative to deplete SP1 in these cells and show loss of ORF75 promoter activity). Finally, they show that ORF75 overexpression activates several viral promoters, including RTA and ORF75 itself. These data suggest new roles in viral latency and gene regulation for this annotated tegument protein and raise interesting new models for both latency and reactivation in KSHV.

Despite the interesting conceptual advance, there are concerns about how the data are presented (e.g., ‘breaking’ of Y-axes rather than converting to log scale) or the lack of error bars. In some cases, the legends suggest technical rather than biological replicates, though it is not easy to tell. Please confirm that all experiments (main and supplemental) have at least 3 biological replicates; this should be stated for blots/gel shifts and represented as error bars in graphs.

Reviewer #2: (No Response)

Reviewer #3: The manuscript by Nair et al. entitled “The Elevated Expression of ORF75, a Lytic KSHV Gene, in Kaposi Sarcoma Lesions is Driven by a GC-rich DNA cis Element in its Promoter Region” explores the elevated expression of KSHV ORF75, a lytic gene, in Kaposi Sarcoma (KS) lesions, where the virus usually maintains latency. Through RNAscope, the authors found that ORF75 RNA is highly expressed in cells positive for latency-associated nuclear antigen (LANA), indicating its prominent role even in latent phases. By employing luciferase reporter assays, they demonstrated that the transcriptional activation of ORF75 is driven by a GC-rich Sp1 cis-element in its promoter, along with distal CCAAT boxes. Sp proteins are shown to interact with these elements, activating the ORF75 promoter in endothelial cells, while a repressive factor in B cells limited its expression in these cells. The study also observed autoregulation, where ORF75 upregulates its expression and other KSHV genes, suggesting a feedback loop during viral replication. Additionally, the accumulation of forms Sp1 during viral latency and lytic phases in PEL cells and endothelial cells hints at a complex regulatory mechanism.

In summary, this work highlights the significant constitutive expression of ORF75 in KS lesions, driven by transcriptional mechanisms involving Sp1 proteins, which could account for its high levels in infected endothelial cells. The data presented in this manuscript convincingly support the hypothesis that ORF75 is highly expressed in Kaposi Sarcoma (KS) lesions due to the activity of a GC-rich Sp1 cis-element in its promoter. The findings regarding Sp1 protein interactions and their role in transcriptional regulation are robust. However, the specificity of Sp protein binding to the promoter elements could be further validated to strengthen the conclusions.

**Part II – Major Issues: Key Experiments Required for Acceptance**

Reviewer #1: 1. Of primary concern are the issues with data presentation:

a. In Figure 2b, the legend indicates 3 time points were taken at day 30, 35, and 40, but the graph does not show multiple timepoints. What timepoint is represented by the 3 data points in each bar of that graph? Also, the graph is presented with a broken y-axis, which appears to have distorted the data above the break as well as the error bar spanning the break for ORF75 in TIME.219 cells. The data should be presented on a log-scale to allow these comparisons without distortion.

b. Figure 2c is missing error bars or lacks 3 biological replicates.

c. Figure S3, panels c-f are either missing error bars or lack 3 biological replicates.

d. In Figure 4a and Figure 5a, for example, the fold changes given are used inconsistently, with those in Figure 5a matching the y-axis value but figure 4a does not. Please correct or clarify.

e. In Figure 4f, no error bars are shown, and the broken y-axis appears to be distorting the data.

f. Figure 8h, data appears distorted above the broken y-axis.

2. The authors are commended for exploring their phenotypes in multiple cell lines, however the frequent cell line switching makes direct comparisons hard, particularly in Fig.6: Fig. 6A makes arguments about ORF75 promoter activity being increased in TIVE, then 6B compares SP1 expression between B and endothelial cells (but use HUVEC, not TIVE), then 6C uses TIME as the endothelial cell representative for gel shifts. Need one endothelial line carried through for direct comparisons.

Reviewer #2: (No Response)

Reviewer #3: 1. The electrophoretic mobility shift assay shows the retardation of the band due to the specific binding of probes to a protein. Since Sp1 binding site was included in the probes, it can be concluded that the retardation of band mobility is due to Sp1 binding. Additionally, using cold competitors to abolish binding suggests that the band is specific to the probe. However, this experiment will benefit from demonstrating the specificity of Sp1 binding to the DNA by utilizing anti-sp1 antibodies to shift the Sp1-Probe complex further. Although the Sp1 mutated probe suggests that the shift in the band mobility is due to the Sp1 binding, having that additional lane will strengthen the data.

2. Figure 5B: The claim of more Sp1 than Sp3 and Sp4 in the nucleus of HUVEC cells seems inaccurate as BCBl-1 cells also show high levels of Sp1 ad compared to Sp3 and Sp4. Fractionation of cytoplasmic and nuclear fractions to demonstrate the proportion of these Sp proteins in these two cell lines, along with the control of accurate fractionation, might provide a better estimation of a proportionate localization.

3. The levels of Sp1 protein through the immune blot in these MA-treated cells used for luciferase assay will show that, indeed, increasing concentrations of MA determine the levels of Sp1.

4. Figure 5D: The use of anti-Sp1 and Sp3 antibodies to show their binding specificity does not convincingly show that the indicated sp1 band is specific to Sp1 binding. Instead, the increasing amounts of anti-Sp1 antibodies in the mix have reduced the band intensity, suggesting that anti-Sp1 antibodies might be competing with the probe binding. Utilizing purified Sp1 and Sp3 proteins for these assays will be more convincing.

5. Figure 5E and F. The levels of Sp1 from exogenously expressing plasmid should be shown to ensure these cells were expressing comparable levels of Sp1.

**Part III – Minor Issues: Editorial and Data Presentation Modifications**

Reviewer #1: 1. The manuscript could be streamlined to remove some of the data that are tangential to the major claims. E.g., data in Fig. S5 and Fig.7 that aren’t quite developed enough to draw clear conclusions.

2. The different binding intensity in the gel shift in Fig6C could be accounted for by modest expression differences of SP1. I would temper the quantitative claims here.

3. It would be clearer to show Figure 6e together with the corresponding data from HEK293T cells currently in Figure 4a.

4. In Figure 8, a supernatant transfer or other measurement of productive reactivation would be useful to know whether ORF75-induced activation of the RTA promoter is sufficient for reactivation into a productive lytic cycle.

5. The authors may consider examining or discussing the expression of K15, which shares an annotated polyA site with ORF75.

6. It could be interesting to speculate in the discussion on potential parallels between their observations of ORF75 and the behavior of EBV ZTA, given their data suggests ORF75 and ORF50 are mutually activating.

7. Typo in ln 508: “a late gene in PEL”

Reviewer #2: (No Response)

Reviewer #3: 1. Figure 3C: The luciferase units seem low unless multiplication factors exist.

2. Figure S3c: These fold changes are not highly significant, mainly when no error bars exist. Error bars should be shown on other panels, as well as other supplemental figures.

3. In the title, the term “cis” should be italicized per convention.

4. Perhaps “lytic” should come after KSHV in the title.

5. Line 28: the term “production” should be “expression” because genes are expressed rather than produced.

6. Line 38: bold parenthesized references

7. Line 51: CRISPR and siRNA techniques are mentioned, but only about knocking out ORF75. Since siRNA is mentioned, perhaps knocking down should be mentioned.

8. Line 81: the word “gene” is missing after “lytic”

9. Line 83: add a space between the period and “To…”

10. Line 85: remove the space after “1A” within the parentheses

11. Line 154: “…within it” is vague and would be improved by explicitly stating “…within the promoter region.”

12. Lines 306, 310, 316, 324: the abbreviation for the cell line used should be SL2, not Sl2.

13. Line 353: “SP1” should be “Sp1” (lowercase “p”)

14. Line 421: “lessor” should be “lesser”

15. Line 426: “access” should be “assess.”

16. Line 455: insert “a” between “that” and “different”

17. Line 536: “It’s…” is always vague and would be improved by explicitly stating, “The lower constitutive expression of the ORF75 promoter…”

18. Line 592: “Bacillus subtilis” should be italicized per convention

19. Line 659: remove the “n” in the word “ration”

20. Line 724: a reference is made to “(Hannah & Davis)” but no year is given, and the citation appears to be missing from the References section

21. Figure 1C: the green dashed-line box appears smaller than the area in the zoomed-in image.

22. Figure 2B: the key indicates that BCBL-1 is represented in blue but in purple.

23. Figure 6C: An asterisk is adjacent to the LANA blots in the WB panel. What is this asterisk supposed to (if anything) indicate?

24. Figure S3B: the x-axis appears to have markers for labels between each cell line dataset. Is this an error, or are there missing labels (and data)?

25. Figure S3C: the “3” is missing in the y-axis label for “pGL3”

26. Figure S4B: This panel does not appear explicitly mentioned in the article's main text, but it would be worth stating that this western blot was performed to verify the expression of Sp proteins.

PLOS authors have the option to publish the peer review history of their article (what does this mean? ). If published, this will include your full peer review and any attached files.

**Do you want your identity to be public for this peer review?** For information about this choice, including consent withdrawal, please see our Privacy Policy .

Reviewer #1: No

Reviewer #2: No

Reviewer #3: No

---

## [Decision Letter · Decision Letter 1]

16 Feb 2025

Dear Dr. Yarchoan,

We are pleased to inform you that your manuscript 'The Elevated Expression of ORF75, a KSHV Lytic Gene, in Kaposi Sarcoma Lesions is Driven by a GC-rich DNA cis Element in its Promoter Region' has been provisionally accepted for publication in PLOS Pathogens.

Best regards,

Dirk P. Dittmer, Ph.D.

Academic Editor

PLOS Pathogens

Robert Kalejta

Section Editor

PLOS Pathogens

Sumita Bhaduri-McIntosh

Editor-in-Chief

PLOS Pathogens

orcid.org/0000-0003-2946-9497

Michael Malim

Editor-in-Chief

PLOS Pathogens

orcid.org/0000-0002-7699-2064

Reviewer Comments (if any, and for reference):

Reviewer's Responses to Questions

**Part I - Summary**

Reviewer #1: The authors have adequately addressed the majority of our prior concerns.

Reviewer #3: The authors have addressed the comments to satisfactory levels.

**Part II – Major Issues: Key Experiments Required for Acceptance**

Reviewer #1: (No Response)

Reviewer #3: None

**Part III – Minor Issues: Editorial and Data Presentation Modifications**

Reviewer #1: In response to our query about whether K15 (which shares a poly(A) site with ORF75) is regulated similarly to ORF75, the authors now present RT-qPCR data tracking K15 and ORF75 mRNA levels (fig S2B). While we agree that ORF75 transcript levels in the 3 cell lines are higher than those of K15, the general trends in expression between cell lines are quite similar for both transcripts, perhaps suggesting that the genes are indeed similarly regulated. So I'm hesitant to agree with their statement on line 116 that these data are "supporting ORF75 116 gene’s independent regulation via its own promoter".

Reviewer #3: None

PLOS authors have the option to publish the peer review history of their article (what does this mean? ). If published, this will include your full peer review and any attached files.

**Do you want your identity to be public for this peer review?** For information about this choice, including consent withdrawal, please see our Privacy Policy .

Reviewer #1: No

Reviewer #3: No

---

## [Editor Report · Acceptance letter]

Dear Dr. Yarchoan,

We are delighted to inform you that your manuscript, "The Elevated Expression of ORF75, a KSHV Lytic Gene, in Kaposi Sarcoma Lesions is Driven by a GC-rich DNA *cis* Element in its Promoter Region ," has been formally accepted for publication in PLOS Pathogens.

Best regards,

Sumita Bhaduri-McIntosh

Editor-in-Chief

PLOS Pathogens

orcid.org/0000-0003-2946-9497

Michael Malim

Editor-in-Chief

PLOS Pathogens

orcid.org/0000-0002-7699-2064